# Control of Slc7a5 sensitivity by the voltage-sensing domain of Kv1 channels

**Shawn M Lamothe[1†], Nazlee Sharmin[2†], Grace Silver[1], Motoyasu Satou[3,4], Yubin Hao[1], Toru Tateno[4], Victoria A Baronas[1], Harley T Kurata[1]***

[1]Department of Pharmacology, Alberta Diabetes Institute, University of Alberta, Edmonton, Canada; [2]School of Dentistry, Faculty of Medicine and Dentistry, University of Alberta, School of Dentistry, Edmonton Clinic Health Academy (ECHA), Edmonton, Canada; [3]Department of Biochemistry, Dokkyo Medical University School of Medicine, Tochigi, Japan; [4]Department of Medicine, Faculty of Medicine and Dentistry, University of Alberta, Edmonton, Canada

**Abstract** Many voltage-dependent ion channels are regulated by accessory proteins. We recently reported powerful regulation of Kv1.2 potassium channels by the amino acid transporter Slc7a5. In this study, we report that Kv1.1 channels are also regulated by Slc7a5, albeit with different functional outcomes. In heterologous expression systems, Kv1.1 exhibits prominent current enhancement ('disinhibition') with holding potentials more negative than $-120$ mV. Knockdown of endogenous Slc7a5 leads to larger Kv1.1 currents and strongly attenuates the disinhibition effect, suggesting that Slc7a5 regulation of Kv1.1 involves channel inhibition that can be reversed by supraphysiological hyperpolarizing voltages. We investigated chimeric combinations of Kv1.1 and Kv1.2, demonstrating that exchange of the voltage-sensing domain controls the sensitivity and response to Slc7a5, and localize a specific position in S1 with prominent effects on Slc7a5 sensitivity. Overall, our study highlights multiple Slc7a5-sensitive Kv1 subunits, and identifies the voltage-sensing domain as a determinant of Slc7a5 modulation of Kv1 channels.

**\*For correspondence:**
kurata@ualberta.ca

[†]These authors contributed equally to this work

**Competing interests:** The authors declare that no competing interests exist.

## Introduction

A wide array of ion channels underlie distinct and regulated patterns of electrical signaling in excitable cells (*Gutman et al., 2005*; *Yu et al., 2005*). Ion channel subtypes possess different voltage dependence, kinetics, sensitivity to signaling cascades, regulation by physiological ions, and other stimuli that contribute to the moment-to-moment and long-term adaptability of electrical signaling in the body. In contrast to the rich complexity of multi-protein complexes known to regulate many synaptic neurotransmitter receptors (*Jacobi and von Engelhardt, 2018*; *Tomita, 2019*), the majority of research on voltage-dependent potassium (Kv) channels has focused on mechanisms of voltage sensitivity. Studies of the *Drosophila* Shaker channel, the first cloned Kv channel, have generated a detailed understanding of core principles of voltage-dependent regulation (*Bezanilla, 2008*; *Bezanilla, 2006*; *Tempel et al., 1988*; *Timpe et al., 1988*). In comparison, regulation of mammalian Kv channels by extrinsic factors such as accessory proteins or signaling cascades is less understood. It is noteworthy that the Kv channels are the most diverse ion channel gene family, with nearly 50 human genes known to encode pore-forming subunits, but there are a relatively small number of recognized and well-studied accessory proteins (*Gutman et al., 2005*).

Based on prior observations of variable Kv1.2 function in different cell types, we hypothesized that Kv1.2 is influenced by a variety of unidentified regulators (*Abraham et al., 2019*; *Baronas et al., 2017*, *Baronas et al., 2016*, *Baronas et al., 2015*; *Rezazadeh et al., 2007*). We have pursued the identification of novel regulatory proteins that may influence this Kv channel, which has served as a structural model for interpretation of functional data (*Long et al., 2007*;

*Matthies et al., 2018*). We identified powerful effects of an amino acid transporter, Slc7a5, on the gating and expression of Kv1.2 (*Baronas et al., 2018*). Slc7a5 has been primarily studied in its role as a transporter of drugs and amino acids (*Barollo et al., 2016*; *Dickens et al., 2017*; *Soares-da-Silva and Serrão, 2004*), and also in the context of nutrient regulation of mTOR signaling (*Nicklin et al., 2009*; *Wolfson et al., 2016*), but is not recognized as a regulator of ion channel function. Recessively inherited Slc7a5 mutations were recently identified in patients with neurological symptoms including autism and motor delay. These traits were attributed to defective Slc7a5-mediated amino acid transport in endothelial cells in the blood–brain barrier, where it is prominently expressed (*Kanai et al., 1998*; *Tărlungeanu et al., 2016*), although Slc7a5 likely has diverse physiological roles in many cell types. For example, low levels of Slc7a5 have been reported in neurons, and other studies have suggested transport-independent functions of Slc7a5 in early development (*Katada and Sakurai, 2019*; *Matsuo et al., 2000*). In addition, Slc7a5 is essential for T cell differentiation and clonal expansion (*Marchingo et al., 2020*), correlated to negative outcomes and growth of many proliferating tumor cell types (*Salisbury and Arthur, 2018*), and causes embryonic lethality in Slc7a5 knockout mice (*Poncet et al., 2020*).

Several structures of Slc7a5 have been recently reported, highlighting its conserved LeuT fold comprising 12 transmembrane helices (*Lee et al., 2019*; *Yan et al., 2019*), but there are few clues into the mechanisms underlying the powerful modulation of Kv1.2 by Slc7a5. Features of Slc7a5-dependent modulation of Kv1.2 currents include a pronounced shift (~−50 mV) of the voltage-dependence of activation, along with prominent inhibition of currents that is relieved by strong negative holding voltages. This voltage-dependent relief of inhibition can be very pronounced, often exceeding 10-fold enhancement of whole cell current. Strikingly, certain epilepsy-linked mutations of Kv1.2 are hypersensitive to Slc7a5-mediated modulation, leading to extraordinarily large (sometimes >100 mV) gating shifts when co-expressed with Slc7a5 (*Baronas et al., 2018*; *Masnada et al., 2017*; *EuroEPINOMICS RES consortium et al., 2015*). Although these effects are very prominent, the underlying structural determinants of the Slc7a5:channel interaction, its specificity among other Kv channels, and the role of Slc7a5 modulation in vivo, remain unclear.

In this study, we expanded our investigation of Slc7a5-mediated regulation to include another prominent neuronal potassium channel, Kv1.1. This channel subunit exhibits overlapping patterns of expression with Kv1.2 in the central nervous system, and often assembles with Kv1.2 into heteromeric channels (*Coleman et al., 1999*; *Manganas and Trimmer, 2000*; *Shamotienko et al., 1997*). Our findings demonstrate that Kv1.1 is especially sensitive to modulation by Slc7a5, with endogenous levels of Slc7a5 having prominent regulatory effects. However, Kv1.1 exhibits different outcomes of Slc7a5 modulation relative to Kv1.2, whereas Kv1.5 is Slc7a5-insensitive. We use these differences to probe for sequence elements that influence Slc7a5 sensitivity, and identify the voltage-sensing domain as an important determinant. In summary, this study highlights a non-canonical regulatory influence of Slc7a5 on multiple Kv1 channels and identifies critical channel segments involved in these effects.

## Results

### Kv1.1 sensitivity to Slc7a5

We recently reported several powerful effects of Slc7a5 on the voltage-gated potassium channel Kv1.2 (*Baronas et al., 2018*). This included a prominent Slc7a5-mediated shift of the voltage-dependence of activation by roughly −50 mV (*Figure 1A*, dashed lines, data reproduced from *Baronas et al., 2018* for comparison). We tested the effects of Slc7a5 co-expression with another prominent neuronal Kv1 channel, Kv1.1. Our initial characterization of Kv1.1 did not reveal a significant shift in the voltage-dependence of activation in the presence of Slc7a5 (*Figure 1A,B*).

A second signature feature of Slc7a5 modulation of Kv1.2 is prominent current enhancement ('disinhibition') when membrane voltage is held at supraphysiological voltages near −120 mV (or more negative). This behavior arises due to an initial Slc7a5-mediated inhibition of channels, which is apparent immediately upon break-in and is then relieved by hyperpolarization. Exemplar records illustrating this relief of Slc7a5-mediated inhibition (*Figure 2A,B*) were generated with intermittent 50 ms depolarizations to +10 mV from a holding potential of −120 mV (note that

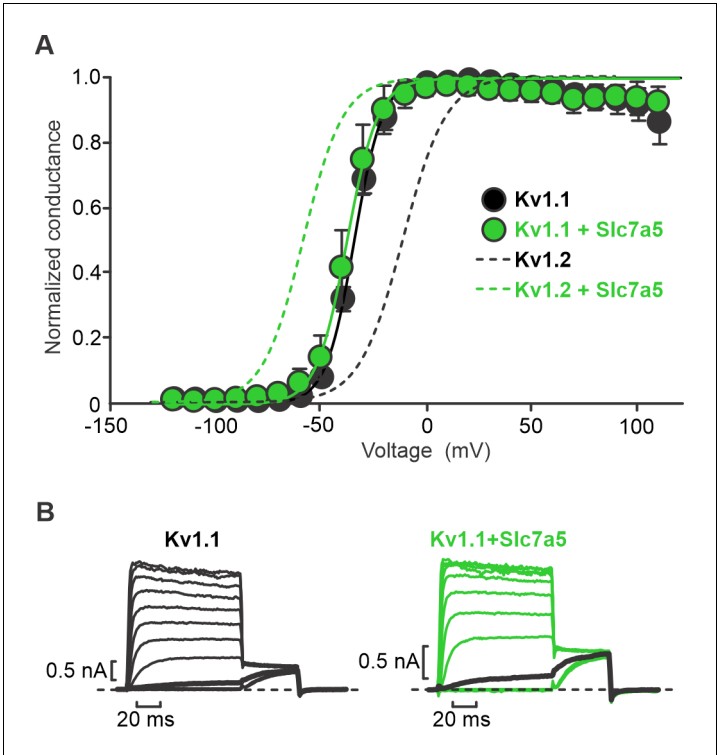

**Figure 1.** Slc7a5 has no effect on voltage-dependent activation of Kv1.1. (**A**) Conductance-voltage relationships were determined for indicated combinations of Kv1.1 and Slc7a5 expressed in LM mouse fibroblasts. Cells were stepped between −130 mV and +110 mV in 10 mV increments, with a tail current voltage of −20 mV. Dashed lines indicate previously reported conductance-voltage relationship in Kv1.2±Slc7a5 (**Baronas et al., 2018**). Fit parameters for Kv1.1 were (co-expression with Slc7a5 in parentheses): $V_{1/2}$ = -34.9 ± 0.3 mV (−37.5 ± 0.2 mV); $k$ = 6.9 ± 0.9 mV (7.3 ± 0.9 mV). No statistical difference in voltage-dependent gating parameters were detected for Kv1.1±Slc7a5. (**B**) Exemplar records illustrating voltage-dependent activation of Kv1.1±Slc7a5 (20 mV interval between voltage steps). Current traces with a −30 mV step are bolded in black.

The online version of this article includes the following source data for figure 1:

**Source data 1.** Slc7a5 effects on voltage-dependence of activation of Kv1.1.

sweeps are concatenated for illustration, the interpulse interval was 2 s). Kv1.2 expressed alone exhibits relatively large currents, and no apparent relief of inhibition (**Figure 2A,C**). However, Kv1.2 co-expression with Slc7a5 (**Figure 2A,D**) leads to baseline currents that are small shortly after whole cell break-in but increase substantially after 20–30 s at a holding potential of −120 mV. We observed an average 5.1 ± 2.2 fold (mean ± S.D.) current enhancement of Kv1.2 with Slc7a5, although this is variable and reached 11.5-fold in some cells (**Figure 2C,D**). In contrast, this effect was prominent for Kv1.1 (**Figure 2B**), even in the absence of co-transfected Slc7a5 (**Figure 2B**), with an average current enhancement of 3.2 ± 1.2 fold (**Figure 2E**). These features (inhibition of baseline current, and relief of inhibition with −120 mV holding potential) became more prominent when Slc7a5 was co-expressed with Kv1.1, with a 6.1 ± 2.6 fold change (**Figure 2B,F**). Taken together, these findings indicate that Slc7a5 causes prominent inhibition of both Kv1.1 and Kv1.2. Reversal of Slc7a5-mediated inhibition by strong hyperpolarizing voltages leads to enhancement of current.

## Knockdown and rescue of Slc7a5-mediated modulation of Kv1.1

Slc7a5 modulation differs between Kv1.1 and Kv1.2, as there is no shift of Kv1.1 activation (**Figure 1**). Also, Kv1.1 channels appear more sensitive to Slc7a5, as they exhibit features of Slc7a5-dependent modulation without co-transfection of Slc7a5 cDNA. Based on these observations, we tested

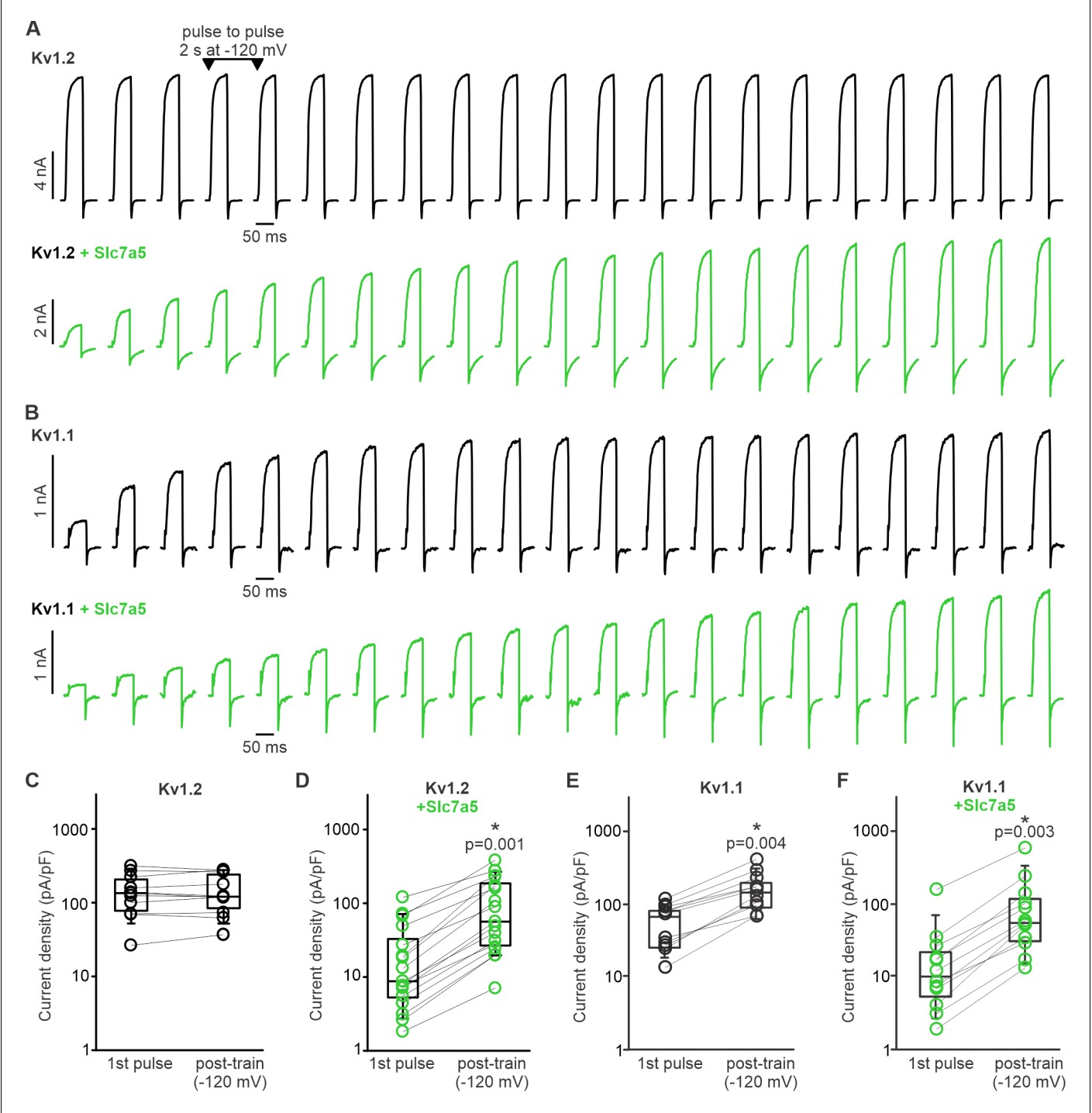

**Figure 2.** Kv1.1 exhibits prominent disinhibition in response to hyperpolarizing (−120 mV) voltage. (A,B) Disinhibition of Kv1.2 (A) or Kv1.1 (B) was tested by delivering repetitive 50 ms depolarizations to +10 mV (every 2 s), with an interpulse holding voltage of −120 mV. When Kv1.2 is expressed alone, currents remain stable during this protocol (A), whereas Kv1.1 exhibits prominent recovery from Slc7a5-mediated inhibition (B). (C–F) Cell-by-cell currents before and after a hyperpolarizing pulse train to −120 mV is illustrated for indicated combinations of Kv1.1, Kv1.2, and Slc7a5 (1st pulse refers to current density of the first +10 mV depolarization, post-train refers to current density of the final +10 mV pulse). Data for Kv1.2±Slc7a5 is reproduced from *Lamothe and Kurata, 2020*. A prominent difference between Kv1.2 and Kv1.1 is that Kv1.1 exhibits disinhibition without a requirement for overexpression of Slc7a5 by co-transfection. Current density pre and post-train was compared using a paired t-test (* indicates p<0.05). Kv1.2 (n = 11, no statistical difference); Kv1.2 + Slc7a5 (n = 16, p=0.001); Kv1.1 (n = 11, p=0.004); Kv1.1 + Slc7a5 (n = 13, p=0.003).

The online version of this article includes the following source data for figure 2:

**Source data 1.** Disinhibition of Kv1.1 at hyperpolarized voltages.

whether Kv1.1 is modulated by endogenous levels of Slc7a5 in LM cells. We generated several Slc7a5 shRNA knockdown cell lines, using lentiviral delivery and puromycin selection to maintain stable shRNA expression. Initial patch clamp recordings from Slc7a5 shRNA cell lines exhibited significant cell-to-cell variability, but many cells exhibited larger baseline current density (immediately after break-in) relative to the parental LM cell line, together with weak current enhancement/disinhibition (*Figure 3—figure supplement 1A,B*). We further isolated individual clonal cell lines by serial dilution of the ShR1 and ShR4 groups, and these clonal cell lines exhibited more consistently attenuated modulation of Kv1.1 (*Figure 3—figure supplement 1C*). Most of the clonal cell lines also exhibited prominent reduction of Slc7a5 protein expression (*Figure 3—figure supplement 1D*). We selected a cell line (ShR4-1) with prominent knockdown of Slc7a5, confirmed by qPCR and Western blot (*Figure 3A–C*).

Expression of Kv1.1 in ShR4-1 cells generated larger currents relative to parental LM cells, and although some modest Kv1.1 current disinhibition persisted in ShR4-1 cells (*Figure 3F*), the magnitude of this effect was significantly attenuated relative to parental LM cells (*Figure 3G*). This was consistent with a loss of Slc7a5-mediated inhibition due to Slc7a5 knockdown. We also used ShR4-1 cells to test whether endogenous Slc7a5 influences voltage-dependent activation of Kv1.1 (which activates at significantly more negative voltages relative to Kv1.2, see *Figure 1*). Knockdown of Slc7a5 in the ShR4-1 cell line did not affect Kv1.1 voltage-dependent activation relative to parental LM cells (*Figure 3—figure supplement 2*), confirming that Kv1.1 and Kv1.2 exhibit distinct functional responses to Slc7a5. The human Slc7a5 transcript is resistant to the mouse-targeted shRNA sequence used for the ShR4-1 cell line, due to two base pair mismatches. Thus, we could rescue Slc7a5 expression in ShR4-1 cells by transfection with human Slc7a5 cDNA (~91% sequence identity, *Figure 3B,E–G*). This rescued Kv1.1 modulation similar to parental LM cells (*Figure 3E*), including suppression of baseline Kv1.1 current density, and prominent current enhancement after holding at −120 mV (*Figure 3F,G*, increased currents between '1st' and 'last' pulses of a −120 mV train). These findings indicate that Slc7a5 is an important contributor to the Kv1.1 modulation observed in LM cells. Additionally, this illustrates that Kv1.1 is particularly sensitive to Slc7a5, leading to modulation by endogenous levels of Slc7a5 in a heterologous cell line.

## Slc7a5 effects on Kv1.2–1.1 heterotetramers

Kv1.2 and Kv1.1 co-assemble to form heteromers (*Al-Sabi et al., 2013*; *Coleman et al., 1999*). Using a concatemeric construct of linked Kv1.2 and Kv1.1 (*Baronas et al., 2015*), we tested Slc7a5 modulation of Kv1.2-Kv1.1 heteromeric channels. Kv1.2-Kv1.1 dimers exhibited a $V_{1/2}$ of −25.5 ± 2.8 mV in parental LM cells and −24.2 ± 3.3 mV in ShR4-1 (*Figure 3—figure supplement 3A*). The linked channel did not undergo any current enhancement in either cell line (*Figure 3—figure supplement 3B–D*) indicating that sensitivity to endogenous Slc7a5 levels is not fully retained in Kv1.2-Kv1.1 heteromers. Overexpression of Slc7a5 in LM and ShR4-1 cells shifted Kv1.2-Kv1.1 channel activation by 30–35 mV, along with pronounced inhibition that was relieved with a −120 mV holding voltage (*Figure 3—figure supplement 3A–D*). We have not determined the stoichiometric requirement for the full Slc7a5-mediated gating shift in Kv1.2 (~−50 mV); however, smaller effects observed in Kv1.2-Kv1.1 linked dimers relative to either homomeric channel suggests that subunit composition influences the functional outcome of Slc7a5 modulation.

## Slc7a5-mediated amino acid transport is not required for Kv1 modulation

It is not known whether Kv1 modulation is due to direct physical interaction with Slc7a5, or via an intermediary or downstream signaling cascade. Slc7a5-mediated amino acid transport is an important contributor to nutrient-dependent activation of mTORC1 (*Nicklin et al., 2009*; *Saxton and Sabatini, 2017*; *Wolfson et al., 2016*, *Wolfson et al., 2016*), so a signal arising from Slc7a5-mediated transport could underlie Kv1 channel modulation (*Figure 4A*). We tested whether pharmacological inhibition of Slc7a5 or mTOR would influence Slc7a5 modulation of Kv1.1. We incubated Kv1.1-transfected LM cells with the Slc7a5 inhibitor BCH (2-amino-bicyclo [2,2,1]heptane-2-carboxylic acid), or the mTOR inhibitor rapamycin (*Figure 4*). Untreated cells exhibited hallmarks of active mTORC1 (*Figure 4B*), in contrast to rapamycin treatment which strongly inhibited basal mTORC1 signaling. This was most obvious as suppression of phospho-S6

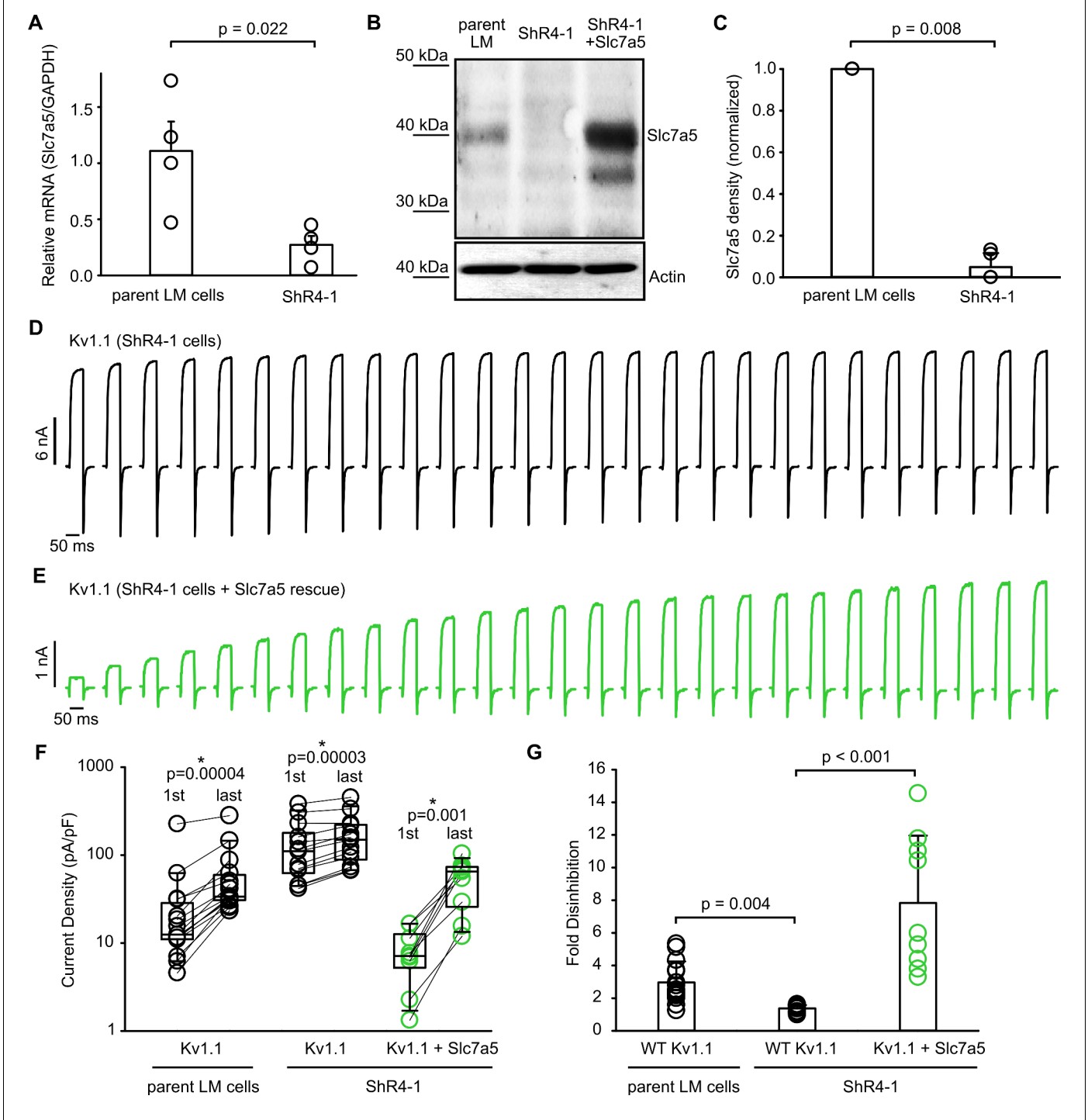

**Figure 3.** Modulation of Kv1.1 function by knockdown and rescue of Slc7a5. (**A**) Quantitative real-time PCR of RNA extracted from parental mouse LM fibroblasts or ShR4-1 (Slc7a5 knockdown cell line) (n = 4, student's t-test). (**B**) Western blot of endogenous Slc7a5 in parental LM cells or ShR4-1 cells. Actin was used as a loading control. (**C**) Densitometry measurements of Slc7a5 expression from parental and ShR4-1 cells (statistical comparison with paired t-test, n = 5). (**D,E**) Exemplar current records illustrating recovery from Slc7a5 inhibition of Kv1.1 during at −120 mV, as described in *Figure 2*, using LM or ShR4-1 cells as indicated. In panel (**E**), Slc7a5 expression is rescued by overexpression with a plasmid encoding human Slc7a5. (**F**) Cell-by-cell currents before and after the −120 mV pulse train of Kv1.1, in parental LM cells or ShR4-1 cell line (n = 9–15, statistical comparison with paired t-test between 1 st pulse and last pulse). (**G**) Fold disinhibition from the first to last pulses of a −120 mV pulse train of Kv1.1 in parental LM cells (mean ± S.D.; 2.96 ± 1.29), ShR4-1 cells (1.36 ± 0.20), or ShR4-1 with Slc7a5 rescue (7.83 ± 4.12)(n = 9–15, Kruskal-Wallis multiple comparisons test, Dunn's post-hoc test).

*Figure 3 continued on next page*

*Figure 3 continued*

The online version of this article includes the following source data and figure supplement(s) for figure 3:

**Source data 1.** Modulation of Kv1.1 by endogenous Slc7a5.
**Figure supplement 1.** Generation of Slc7a5 knockdown cell lines.
**Figure supplement 2.** Knockdown of Slc7a5 does not alter voltage-dependent activation of Kv1.1.
**Figure supplement 3.** Slc7a5 suppresses current and shifts the voltage-dependence of activation of Kv1.2–1.1 heterotetramers.

(p-S6), but also apparent in reduced levels of phospho-mTOR, and the appearance of lower molecular weight forms (likely due to dephosphorylation) of 4-EBP1 (*Figure 4B*). However, after rapamycin treatment, electrophysiological features of Slc7a5 modulation of Kv1.1 remained prominent (*Figure 4C*). There was no alteration of baseline Kv1.1 current levels, and prominent relief of inhibition of Kv1.1 was observed (*Figure 4C*, increased currents between the 1st and last pulses of a −120 mV train). In LM or ShR4-1 cells, pharmacological suppression (by BCH) or shRNA knockdown of Slc7a5 did not influence p-S6 abundance or other markers of mTORC1 signaling (*Figure 4—figure supplement 1A*). However, even though Slc7a5 knockdown clearly attenuates Kv1.1 modulation (*Figure 3*), Slc7a5-mediated gating effects were resistant to pharmacological inhibition by BCH (*Figure 4C*). Lastly, BCH or rapamycin did not influence the voltage-dependence of activation of Kv1.1 (*Figure 4—figure supplement 1B–D*).

Investigation of Slc7a5 modulation of Kv channels was originally prompted by mass spectrometry identification of proteins in proximity to Kv1.2, and we previously demonstrated proximity of heterologously expressed Kv1.2 and Slc7a5 using a BRET assay (*Baronas et al., 2018*). We also detect a BRET signal between heterologously expressed Kv1.1 and Slc7a5 (*Figure 4D–F*). Since Kv1.1 can assemble as a homotetramer, co-expression of Kv1.1-nanoluc (bioluminescent donor) with EGFP-Kv1.1 (acceptor) generates an emission with a peak wavelength of 510–520 nm, consistent with excitation of the EGFP tag by nanoluc. Although not as large as the EGFP-Kv1.1-positive control, EGFP-Slc7a5 also generated a consistent BRET signal when co-expressed with Kv1.1-nanoluc. In contrast, the closely related transporter Slc7a6 did not generate a discernible BRET signal (*Figure 4F*).

Taken together, *Figure 4* illustrates that Slc7a5 is likely in close proximity to Kv1.1, and that amino acid transport or mTORC1 activation downstream of Slc7a5 are not required for modulation of Kv1 channels. It should be noted that we have not consistently observed co-immunoprecipitation of Kv1.1 and Slc7a5, suggesting that the association is detergent sensitive or may involve additional unknown proteins. Also, we have not explicitly ruled out involvement of mTORC2, and there are variable outcomes of mTORC2 inhibition by rapamycin in different cell types (*Sarbassov et al., 2006*). However, mTORC1 is considered to be the primary amino-acid-responsive mTOR complex that would be expected to be modulated by Slc7a5 (*Saxton and Sabatini, 2017*).

## Slc7a5-mediated inhibition is distinct from C-type inactivation

Slc7a5-mediated inhibition of Kv1.1 and Kv1.2 traps channels in a non-conducting state and can be relieved by supraphysiological hyperpolarizing voltages (*Figures 2–4*). We previously demonstrated that mutations that enhance susceptibility to C-type inactivation in Kv1.2 could alter Slc7a5 modulation (*Baronas et al., 2018*; *Lamothe and Kurata, 2020*), although the detailed molecular mechanism of Slc7a5-mediated inhibition has remained unclear. We investigated the relationship between C-type inactivation and Slc7a5-mediated inhibition of Kv1.1. In ShR4-1 cells transfected with Kv1.1, currents exhibit very slow inactivation, and Slc7a5 has no consistent effect on the extent of inactivation observed during 5 s depolarizing pulses (+40 mV, *Figure 5A,B*). Building on previous experiments demonstrating that Kv1.1 modulation persists in the presence of the Slc7a5 inhibitor BCH (*Figure 4*), we also tested a transport-deficient Slc7a5[F252A] mutant (*Singh and Ecker, 2018*), which also has no consistent effect on Kv1.1 inactivation (*Figure 5A,B*). In Kv1.2, prominent acceleration of inactivation by Slc7a5 was apparent in Kv1.2[V381T] mutant channels, equivalent to *Shaker* position T449, which influences susceptibility to C-type inactivation in *Shaker* (*López-Barneo et al., 1993*) and Kv1.2 (*Goodchild et al., 2012*). We tested inactivation of the analogous mutation Kv1.1 [Y379T] with Slc7a5 and Slc7a5[F252A] (*Figure 5—figure supplement 1*), and observed that Slc7a5

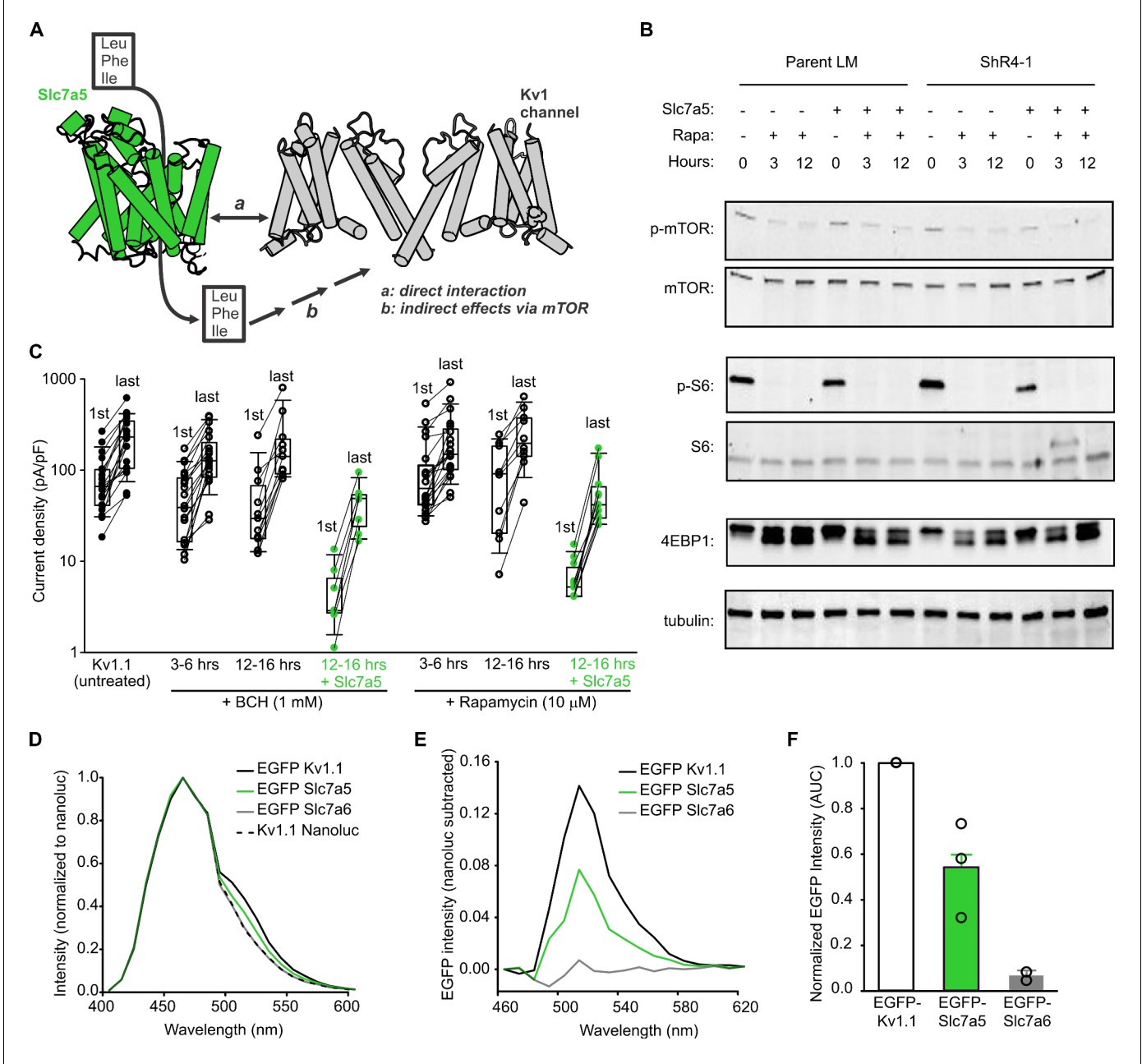

**Figure 4.** Slc7a5-mediated inhibition of Kv1.1 is resistant to suppression of the Slc7a5-mTORC1 signaling axis. (**A**) Schematic model depicting distinct possibilities for Slc7a5 modulation of Kv1 channels via (**a**) direct interaction or (**b**) indirect effects arising from amino acid activation of mTOR. (**B**) Western blot detection of markers of mTOR activation, including total mTOR, phospho-mTOR, total S6, phospho-S6, total 4EBP, in parental LM or ShR4-1 cells treated with 10 μM rapamycin (Rapa) and/or Slc7a5 overexpression, as indicated. β-tubulin was used as a loading control. (**C**) Cell by cell current density before and after a −120 mV pulse train (30 s) is illustrated for Kv1.1 channels ± Slc7a5 (LM cells) treated with 1 mM BCH, or 10 μM rapamycin as indicated. (**D**) Emission spectra were collected from LM cells transfected with indicated combinations of Kv1.1-nanoluc, EGFP-Kv1.1, EGFP-Slc7a5, or EGFP-Slc7a6. (**E**) EGFP spectra (nanoluc-subtracted) were measured for the indicated EGFP-tagged acceptors co-expressed with Kv1.1-nanoluc. (**F**) Area under the curve (AUC) for each BRET acceptor in (**E**) was normalized to the positive control AUC (Kv1.1-nanoluc + EGFP-Kv1.1). The online version of this article includes the following source data and figure supplement(s) for figure 4:

**Source data 1.** Rapamycin and BCH effects on Slc7a5 modulation of Kv1.1.
**Figure supplement 1.** Slc7a5 or mTORC1 inhibition does not influence voltage-dependent gating of Kv1.1.

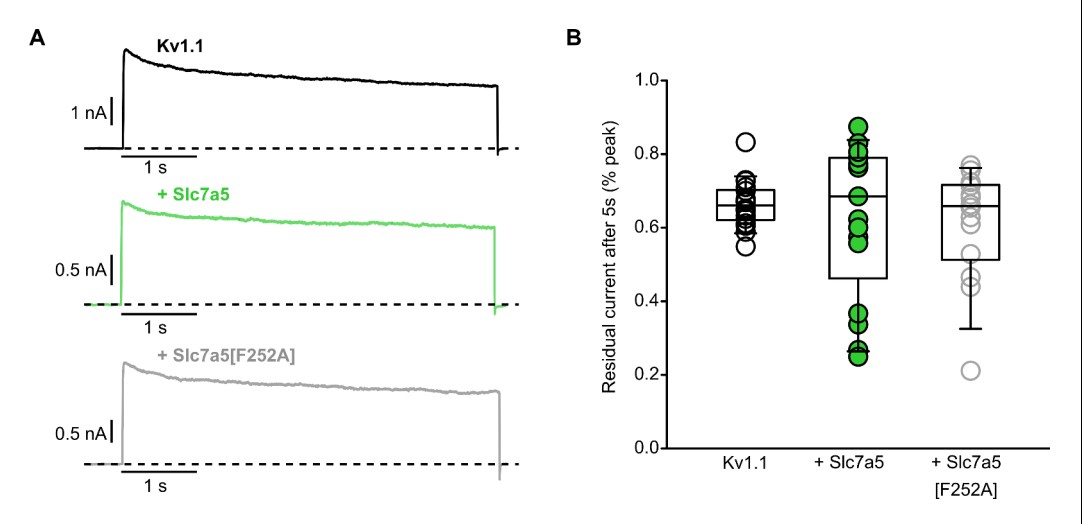

**Figure 5.** Slc7a5 does not accelerate WT Kv1.1 inactivation. (**A**) WT Kv1.1 channels were expressed in mouse LM cells with Slc7a5 or Slc7a5[F252A], as indicated. Exemplar current traces illustrate inactivation elicited by depolarization to 40 mV for 5 s (−80 mV holding potential). Prior to this long depolarization, currents have been disinhibited with a −120 mV holding voltage, as described in *Figure 2*. (**B**) Cell-by-cell inactivation of WT Kv1.1±Slc7a5 and Slc7a5[F252A]. Current amplitude after 5 s was normalized to peak current (% of peak) on a cell-by-cell basis. No statistical difference was detected in the % residual current between all three groups.

The online version of this article includes the following source data and figure supplement(s) for figure 5:

**Source data 1.** Modulation of Kv1.1 inactivation by Slc7a5.

**Figure supplement 1.** Slc7a5 enhances inactivation of Kv1.1[Y379T] channels.

and Slc7a5[F252A] enhance the rate of inactivation, although this effect exhibited significant cell-to-cell variability (*Figure 5—figure supplement 1C*). We suspect this variation arises from variable expression of Slc7a5, although there may also be other regulatory factors that influence the response of Kv1.1 to Slc7a5. The absence of effects of Slc7a5 on inactivation of WT Kv1.1 suggests that accelerated C-type inactivation does not underlie Slc7a5-mediated inhibition. However, the effects of the Kv1.1[Y379T] mutation suggest that there may be some interplay between C-type inactivation and Slc7a5-mediated inhibition. It is also noteworthy that effects of Slc7a5 on the inactivation rate (and the $V_{1/2}$ of activation of Kv1.2 as reported in *Baronas et al., 2018*) remain intact even after relieving Slc7a5-mediated inhibition of currents. We interpret this to indicate that strong hyperpolarizing stimuli allow channels to recover from an inhibited state, but do not 'break apart' the complex that underlies Slc7a5 modulation.

To further explore the mechanism of Slc7a5 inhibition of Kv1.1, we compared recovery from Slc7a5-mediated inhibition versus inactivation (*Figure 6*). Exemplar currents (*Figure 6A*, *left*) illustrate recovery from Slc7a5-mediated inhibition in a sequence of three depolarizations to +10 mV: point 'a' shortly after whole cell break-in, point 'b' after 30 s holding at −80 mV, and point 'c' after 30 s holding at −120 mV. In the absence of Slc7a5, Kv1.1 channels exhibit large currents at point 'a'. As there is very little initial Slc7a5-mediated inhibition, very little relief is observed at holding voltages of either −80 mV or −120 mV. However, co-expression of Slc7a5 or Slc7a5[F252A] strongly suppressed initial currents (*Figure 6A*, *left*). A holding voltage of −80 mV did not effectively rescue this inhibition, but a holding voltage of −120 mV prominently recovered currents (summarized cell-by-cell in *Figure 6B*). These data recapitulate the relief of inhibition effect described in *Figures 2* and *3*, but further illustrate the requirement for repolarization to −120 mV.

Following hyperpolarizations to relieve Slc7a5-mediated inhibition, channels were inactivated with 5 s pulses to +40 mV, and held at −80 mV to recover from inactivation (exemplar currents in *Figure 6A*, *right* panels). In all conditions, Kv1.1 recovers from inactivation with a holding voltage of −80 mV, as currents measured at point 'e' are comparable to point 'd' (*Figure 6C*). This finding suggests that Slc7a5-induced inhibition is not equivalent to C-type inactivation, because different

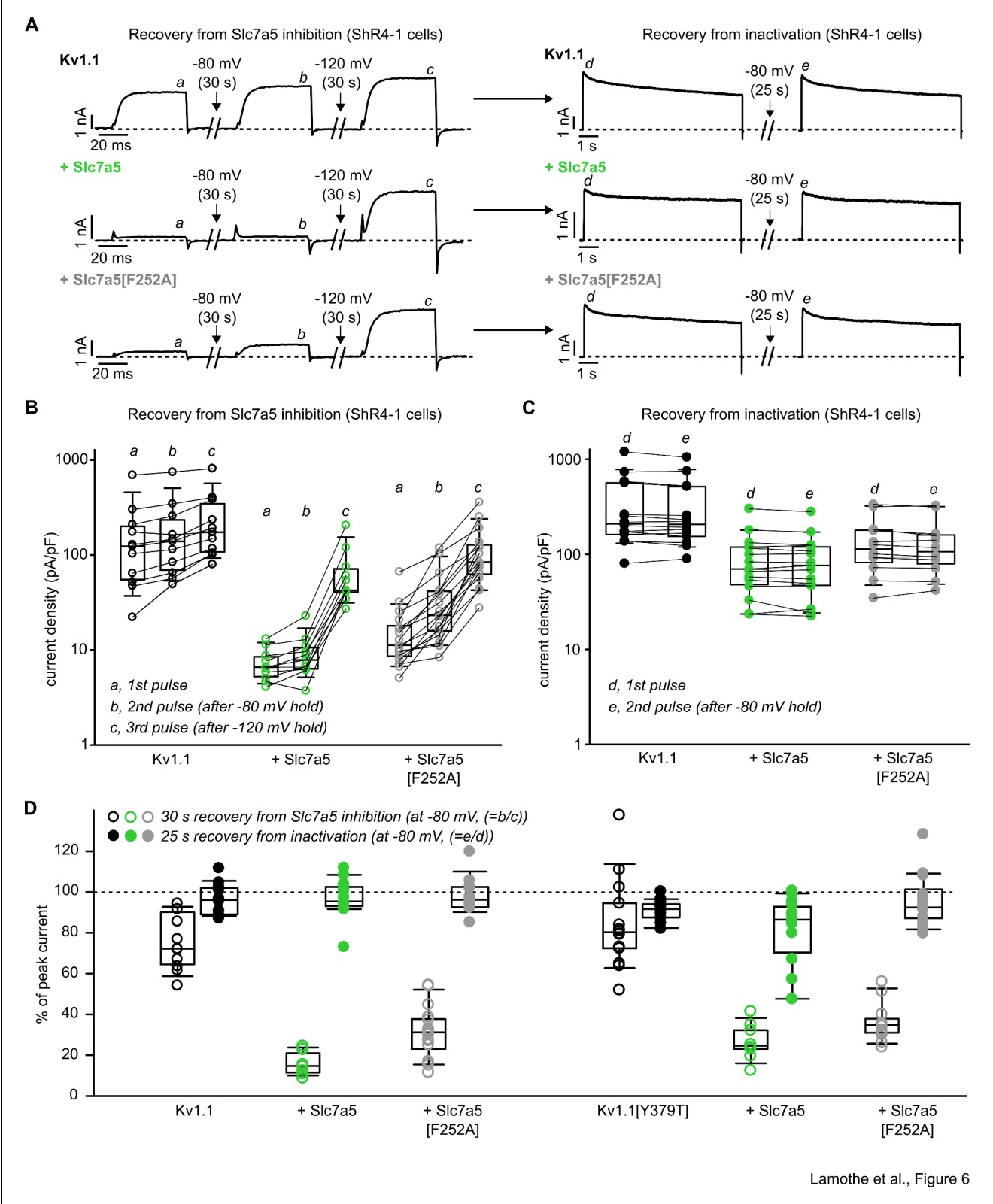

**Figure 6.** Distinct recovery from Slc7a5 inhibition versus inactivation of Kv1.1. (**A**) Exemplar patch clamp recordings of Kv1.1 and Slc7a5 combinations as indicated, in ShR4-1 cells. *Left panel* illustrates recovery from Slc7a5 inhibition. Currents were recorded at 10 mV after break-in, after 30 s at −80 mV holding voltage, and after 30 s at −120 mV. *Right panel* illustrates recovery from inactivation (measured after the recovery from Slc7a5 inhibition shown in the left panel). Cells were depolarized to 40 mV for 5 s, held at a −80 mV recovery potential (25 s) followed by a 2nd pulse to 40 mV. (**B**) Cell-by-cell

*Figure 6 continued on next page*

*Figure 6 continued*

changes in current density (at +10 mV) in response to recovery from Slc7a5 (depicted in panel A, left). Currents were measured where indicated by a, b, c, in panel A (Kv1.1, n = 11; Kv1.1 + Slc7a5, n = 11; Kv1.1 + Slc7a5[F252A], n = 17). (C) Cell-by-cell changes in peak current density in response to recovery from inactivation (depicted in panel A, *right*). Currents were measured where indicated by d, e, in panel A (Kv1.1, n = 14; Kv1.1 + Slc7a5, n = 15; Kv1.1 + Slc7a5[F252A], n = 17). (D) Percentage of peak current achieved during the −80 mV disinhibition protocol (calculated as b/c in panel A) versus the −80 mV recovery from inactivation protocol (calculated as e/d). See Figure supplement for additional details on Kv1.1[Y379T].

The online version of this article includes the following source data and figure supplement(s) for figure 6:

**Source data 1.** Distinct voltage-dependence of recovery from inactivation versus Slc7a5-mediated inhibition.
**Figure supplement 1.** Distinct recovery from Slc7a5 inhibition versus inactivation of Kv1.1[Y379T].

voltages are required for recovery from these different conditions. To test this more stringently, we carried out similar experiments using Kv1.1[Y379T] (*Figure 6—figure supplement 1*). Although Kv1.1[Y379T] exhibits far more prominent inactivation than WT Kv1.1, a similar outcome was observed. Strong hyperpolarization to −120 mV was required to rescue currents from Slc7a5 inhibition after break-in (*Figure 6—figure supplement 1B*). However, once this inhibition was relieved, channels could recover from nearly complete inactivation with a holding potential of −80 mV (*Figure 6—figure supplement 1A,C*). The disparity in the voltage-dependence of recovery from inactivation versus Slc7a5-mediated inhibition is summarized in *Figure 6D*, highlighting that nearly complete recovery from C-type inactivation (solid symbols, ratio of points e/d) is observed with a holding voltage of −80 mV, whereas this voltage does not recover Slc7a5-mediated inhibition (open symbols, ratio of points b/c). These findings indicate that Slc7a5-mediated inhibition stabilizes channels by a mechanism distinct from C-type inactivation. Nevertheless, there appears to be some interaction between these processes because Slc7a5 can influence the inactivation rate in channel mutants that are prone to C-type inactivation (*Figure 5—figure supplement 1*; *Baronas et al., 2018*).

## Slc7a5 sensitivity is controlled by the voltage-sensing domain

Our findings illustrate that Kv1.1 is sensitive to endogenous levels of Slc7a5, but does not exhibit a prominent gating shift. In contrast, Kv1.2 requires higher expression of Slc7a5 to observe an effect (i.e. it is less sensitive to Slc7a5), but it exhibits a 'signature' −50 mV shift of voltage-dependent activation. Additionally, we have reported that Kv1.5 exhibits no hallmarks of Slc7a5 modulation (i.e. no voltage shift or disinhibition; *Baronas et al., 2018*). We used these differences to investigate structural determinants of Slc7a5 sensitivity. Our first approach involved chimeras of Kv1.2 and Kv1.1 (*Figure 7*). For each chimera, Slc7a5 effects on current magnitude before and after a −120 mV holding voltage is illustrated in *Figure 7A*, and effects on voltage-dependent gating are presented in *Figure 7B–E*. The N-terminus, S1, and S2 segments of Kv1.2 can be replaced with Kv1.1 sequence, while still preserving a large Slc7a5-mediated shift in voltage-dependent gating (*Figure 7A–D*). Further replacement of the S3 and S4 segments (and a small portion of the pore) in the Kv1.1S5/Kv1.2 chimera led to a significant switch toward a Kv1.1-like phenotype (*Figure 7A,E*), with a minimal Slc7a5-mediated gating shift and prominent current enhancement, even in the absence of transfected Slc7a5.

We also tested more subtle chimeric replacements of segments of the Kv1.2 voltage sensor to potentially map these effects more precisely. The only sequence differences between the Kv1.1S2/Kv1.2 and the Kv1.1S5/Kv1.2 chimeras were in the S3-S4 linker, and residue I257 in S4. We replaced the S3-S4 linker of Kv1.2 with the corresponding sequence from Kv1.1 (Kv1.2(1.1-S3/S4)), and also tested the I257F mutation. However, neither of these chimeras/mutants caused Kv1.2 to switch to a Kv1.1-like response (*Figure 7—figure supplement 1A–C*). These findings suggest that sequence differences in isolated segments of the voltage sensor do not account for the different functional effects of Slc7a5 on Kv1.1 vs. Kv1.2.

## VSD chimeras swap prominent features of Slc7a5 sensitivity

We also swapped the entire S1-S4 segments of Kv1.1 and Kv1.2 (*Figure 7—figure supplement 2*). Introduction of the voltage-sensing domain of Kv1.2 into Kv1.1 (Kv1.2VSD/Kv1.1) transferred

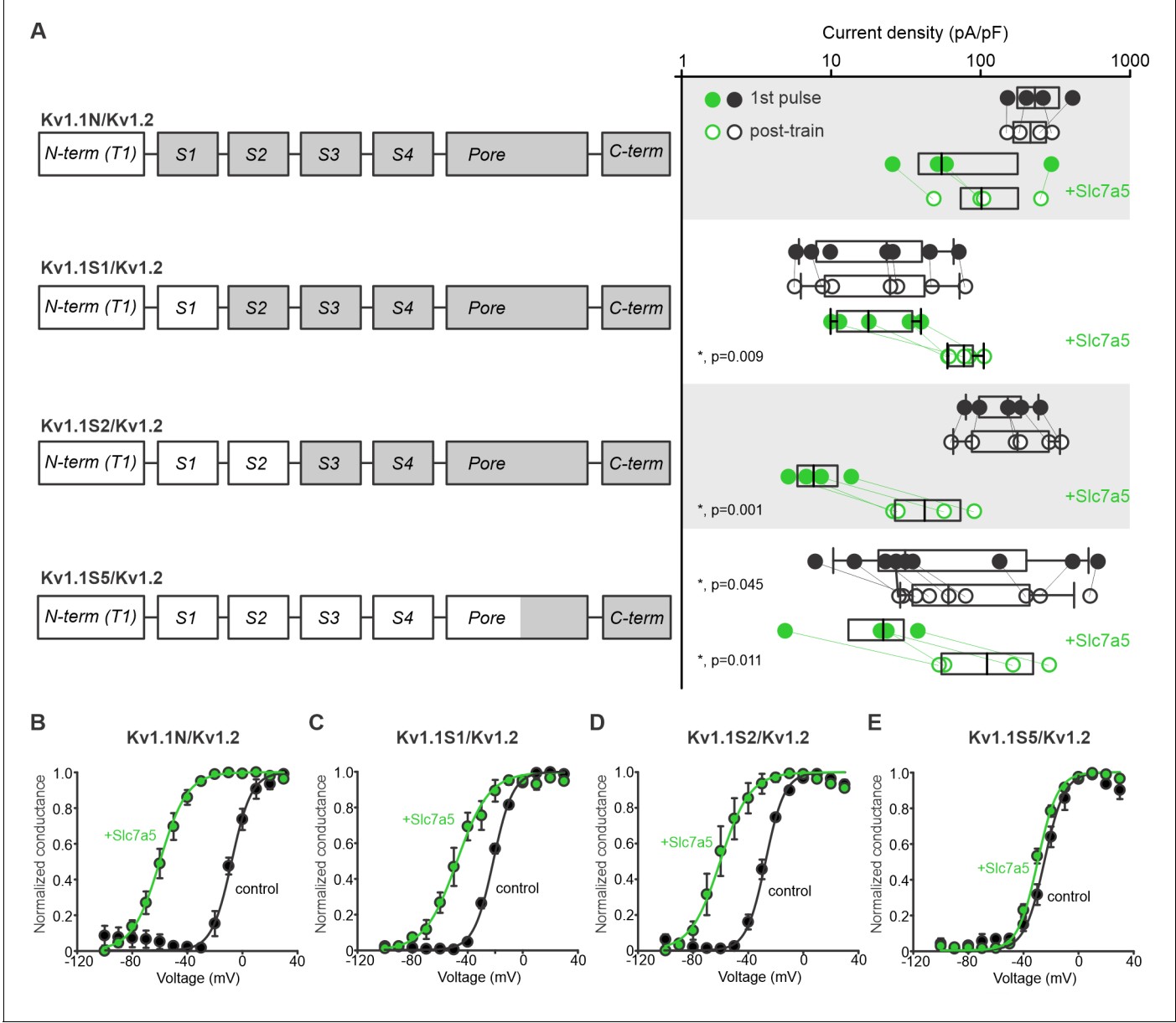

**Figure 7.** Chimeric analysis of Kv1.1 and Kv1.2 sensitivity to Slc7a5-mediated disinhibition and shifts of voltage-dependent activation. (**A**) Cartoons illustrate chimeric channel design, in which increasing segments of Kv1.1 (white) were introduced into Kv1.2 (grey), beginning with the N-terminus. Current disinhibition by a hyperpolarizing train to −120 mV was assessed as described in *Figure 2*, in the presence or absence of Slc7a5. (**B–E**) Conductance-voltage relationships were measured for all chimeric channels, in the presence and absence of Slc7a5. Gating parameters (+Slc7a5 in parentheses) for Kv1.1N/Kv1.2 were: $V_{1/2}$ = -8.7 ± 2 mV (−61 ± 3 mV); $k$ = 7 ± 1 mV (8.3 ± 0.5 mV); for Kv1.1S1/Kv1.2: $V_{1/2}$ = -21.5 ± 0.7 mV (−42 ± 6 mV); $k$ = 7.1 ± 0.2 mV (9 ± 1 mV), for Kv1.1S2/Kv1.2: $V_{1/2}$ = -30.1 ± 0.4 mV (−53 ± 8 mV); $k$ = 7.4 ± 0.2 mV (8.2 ± 0.4 mV), and for Kv1.1S5/Kv1.2: $V_{1/2}$ = −25 ± 2 mV (−30 ± 2 mV); $k$ = 7.4 ± 0.3 mV (7.2 ± 0.3 mV). Prominent shifts in voltage-dependent gating were observed in all chimeras except the Kv1.1S5/Kv1.2, comprising primarily the transmembrane domains of Kv1.1.

The online version of this article includes the following source data and figure supplement(s) for figure 7:

**Source data 1.** Voltage sensor modulation of Slc7a5 sensitivity of Kv1.1 and Kv1.2.

**Figure supplement 1.** Slc7a5-mediated gating shift is preserved in I257 or S3-S4 linker mutants of Kv1.2.

**Figure supplement 2.** The voltage-sensing domain influences Slc7a5 sensitivity and response.

characteristic features of Kv1.2 modulation by Slc7a5. These chimeric channels exhibited a prominent Slc7a5-dependent shift in voltage-dependent gating, and current disinhibition, comparable to Kv1.2 (*Figure 7—figure supplement 2A,B*). In addition, these features of Slc7a5 modulation were absent when the Kv1.2VSD/Kv1.1 chimera was expressed alone (*Figure 7—figure supplement 2A, B*). The complimentary chimera with the voltage-sensing domain of Kv1.1 transplanted into Kv1.2 (Kv1.1VSD/Kv1.2) did not have such clear cut effects, but altered the Slc7a5 sensitivity of Kv1.2 (*Figure 7—figure supplement 2C,D*). The Kv1.1VSD/Kv1.2 chimera exhibited prominent disinhibition of current even in the absence of Slc7a5, and an attenuated shift in voltage dependence of activation when co-expressed with Slc7a5 (*Figure 7—figure supplement 2C,D*). Overall, although these findings illustrate the importance of the VSD, they reinforce our finding (*Figure 7—figure supplement 1*) that this approach does not reveal a clear structural determinant of the different Slc7a5 responses of Kv1.1 and Kv1.2. This may be an inherent shortcoming of this chimeric approach, as both Kv1.1 and Kv1.2 are sensitive to Slc7a5.

## Precise mapping of determinants of Slc7a5 sensitivity

In order to more clearly pinpoint regions of the VSD involved in Slc7a5 sensitivity, we used a chimeric strategy with Kv1.2 and Kv1.5. We believe this was a more useful approach because Kv1.5 is not sensitive to Slc7a5. The initial chimeric design is illustrated in *Figure 8—figure supplement 1A*. Although Slc7a5-mediated gating shifts were sometimes variable, we observed that channels became clearly resistant to Slc7a5-mediated gating shifts after the S1-S2 transmembrane helices of Kv1.2 were replaced with the sequence from Kv1.5 (*Figure 8—figure supplement 1B*).

We generated additional chimeric channels in which small segments of the S1-S2 region of Kv1.2 were replaced with corresponding segments of Kv1.5. We generated conductance-voltage relationships for each of these chimeric channels (*Figure 8—figure supplement 2A–D*). Swapping the S1-S2 linker, S2 segment, or S2-S3 linker, had no consistent effect on the Slc7a5-mediated shift of Kv1.2 gating. In contrast, replacing the S1 transmembrane segment of Kv1.2 with Kv1.5 sequence strongly attenuated the Slc7a5-mediated gating shift (*Figure 8—figure supplement 2E*).

## Point mutations in S1 influence Slc7a5 sensitivity

We extended these chimeric analyses with point mutants based on five amino acid differences in S1 of Kv1.2 and Kv1.5 (*Figure 8A*). Conductance-voltage relationships for each mutant (*Figure 8C–G*) and a summary of $V_{1/2}$ of activation (*Figure 8B*) reveal that most point mutants did not weaken Slc7a5-mediated effects. However, the Kv1.2[I164A] mutation near the intracellular boundary of S1 prominently attenuated the Slc7a5-mediated gating shift. (*Figure 8B–G*). This highlights a single amino acid position that strongly influences Slc7a5 sensitivity of Kv1 channels.

We investigated whether a role for this position would extend/generalize to Slc7a5-mediated inhibition of Kv1.1. We mutated the equivalent V168 position in Kv1.1 to I or A (from Kv1.2 or Kv1.5, respectively, *Figure 9A*), and tested for hallmarks of Slc7a5 modulation (*Figure 9B*). Recordings in parental LM cells (with Slc7a5 modulation intact) illustrate that Kv1.1[V168I] (gray symbols) retains Slc7a5 sensitivity, as these channels exhibited suppressed currents upon break-in ('1st pulse'), along with prominent enhancement after holding at −120 mV ('last' pulse). In contrast, Kv1.1[V168A] (green) exhibited large basal currents and comparably little current enhancement, suggesting weaker sensitivity to Slc7a5 (*Figure 9B*, summarized in lower panel). In ShR4-1 cells, larger currents and negligible disinhibition were observed in all cases (*Figure 9B*), further emphasizing that Slc7a5 strongly influences the outcome observed for Kv1.1[V168] mutations. An alternative visualization of these data (*Figure 9C,D*) illustrates cell-by-cell the relationship between baseline currents (i.e. 1st pulse immediately after break-in) and disinhibition of current arising from a −120 mV holding potential. In LM cells (*Figure 9C*), WT or Kv1.1[V168I] currents are typically small immediately after break-in, but exhibit prominent recovery from Slc7a5-mediated inhibition (black and grey). In contrast, Kv1.1[V168A] (green) exhibits large currents and very modest disinhibition (illustrating weak inhibition by Slc7a5). In ShR4-1 cells (*Figure 9D*), effects of V168 mutations are blunted, illustrating that outcomes of V168 mutants are strongly influenced by the presence of Slc7a5, rather than effects on intrinsic channel function.

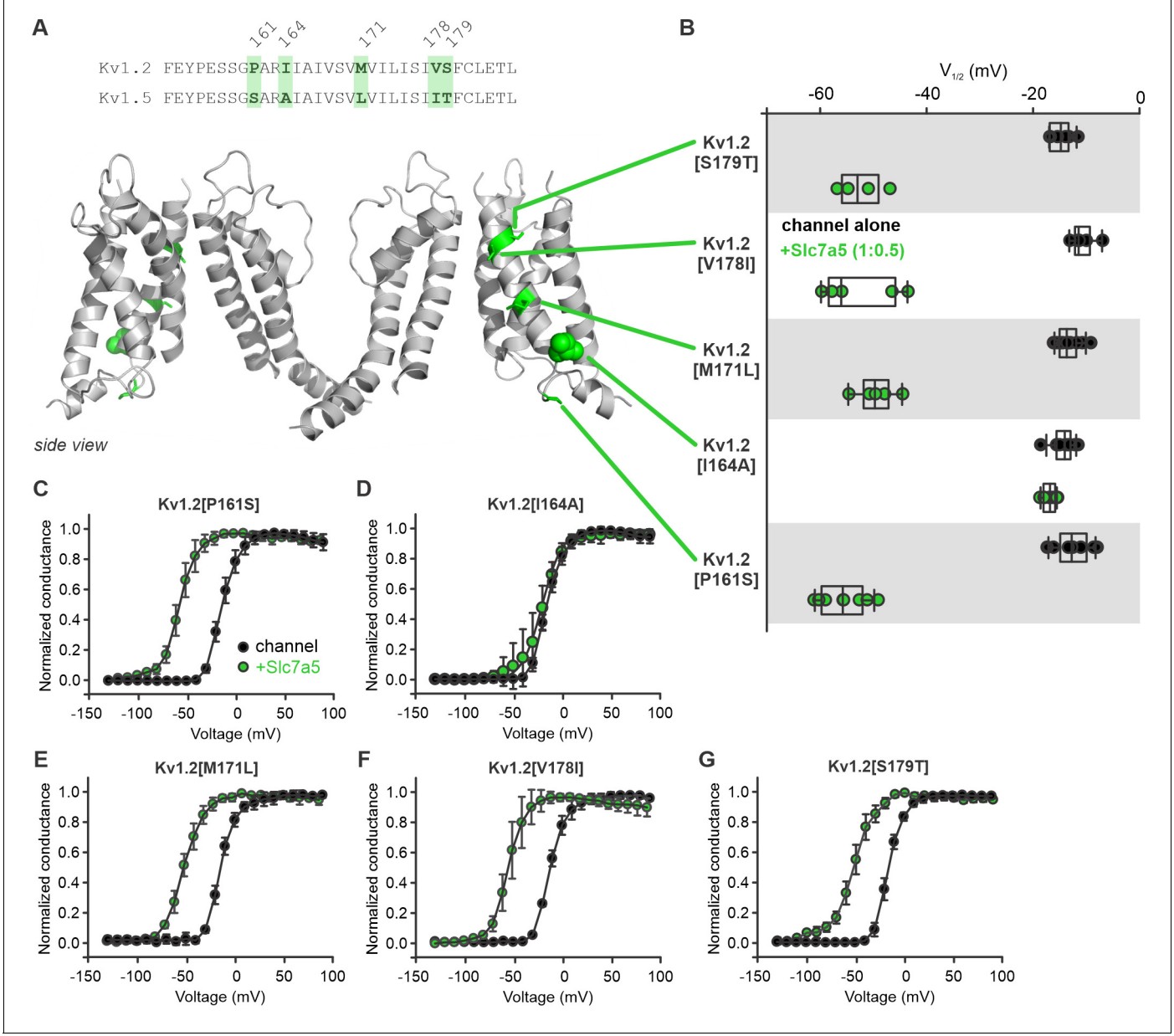

**Figure 8.** Amino acid residue I164 influences Kv1.2 sensitivity to Slc7a5 modulation. (A) *Top*, amino acid sequence comparison of the S1 domain between Kv1.2 and Kv1.5. Different amino acid residues between the two channels are highlighted in green. *Bottom*, Schematic model (side view) of the Kv1.2 channel with dissimilar amino acids from Kv1.5 highlighted in green. (B) Voltage at half activation ($V_{1/2}$) plots of individual cells of Kv1.2 point mutants in the S1 domain substituted with the corresponding amino acid in Kv1.5. The Kv1.2 point mutants were recorded and analyzed in the presence or absence of Slc7a5. (C–G) Conductance-voltage relationships of the Kv1.2 point mutants were measured in the presence or absence of Slc7a5. Fit parameters were (mean ± S.D., Slc7a5 in parentheses): P161S, $V_{1/2}$ = -12.6 ± 3.0 mV (−55.7 ± 4.4 mV), k = 8.7 ± 1.4 mV (10 ± 2.5 mV); I164A, $V_{1/2}$ = -14.4 ± 2.1 mV (−16.9 ± 1.2 mV), k = 8.4 ± 1.3 mV (11.0 ± 3.9 mV); M171L, $V_{1/2}$ = -8.6 ± 2.1 mV (−49.5 ± 3.7 mV), k = 8.7 ± 1.4 mV (10.8 ± 1.2 mV); V178I, $V_{1/2}$ = -8.9 ± 2.3 mV (−52.8 ± 7.2 mV), k = 8.8 ± 1.8 mV (9.2 ± 2.6 mV); S179T, $V_{1/2}$ = -11.9 ± 2.0 mV (−52.5 ± 4.4 mV), k = 7.8 ± 0.5 mV (−11.9 ± 0.2 mV).

The online version of this article includes the following source data and figure supplement(s) for figure 8:

**Source data 1.** Voltage sensor modulation of Slc7a5 sensitivity of Kv1.2 and Kv1.5.

**Figure supplement 1.** Chimeric analysis of Kv1.5 and Kv1.2 sensitivity to Slc7a5-mediated shifts of voltage-dependent activation.

**Figure supplement 2.** Kv1.2 sensitivity to Slc7a5 is localized to the S1 transmembrane segment.

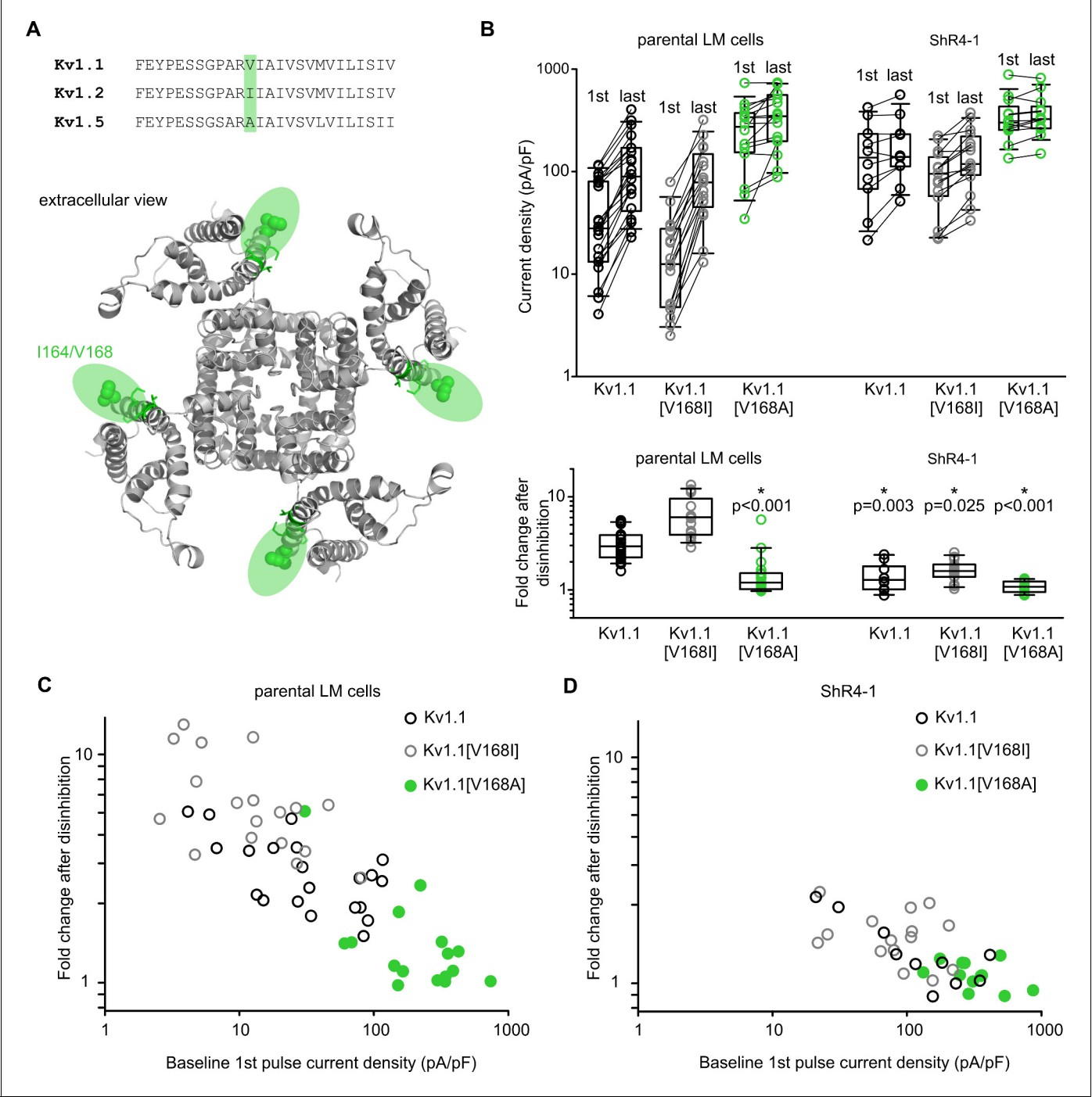

**Figure 9.** Altered Slc7a5 modulation of Kv1.1 V168 mutations. (**A**) *Top*, amino acid sequence comparison of S1 in Kv1.1, Kv1.2 and Kv1.5. Sequence variation at Kv1.1 position V168 is highlighted in green (I164 in Kv1.2, A251 in Kv1.5). *Bottom*, structural model of Kv1.2 channel highlighting amino acid position I164 (equivalent to Kv1.1 V168). (**B**) Cell-by-cell current density before and after a −120 mV pulse train of indicated channel mutants, in the parental LM cells or ShR4-1 cell line. *Lower panel*, fold disinhibition for Kv1.1 S1 mutant channels (fold disinhibition = last pulse/1st pulse) after a −120 mV pulse train. (**C and D**) Cell-by-cell correlation of baseline (1st pulse) current density and fold disinhibition after a −120 mV pulse train, for Kv1.1 mutants and cell lines as indicated.

The online version of this article includes the following source data for figure 9:

**Source data 1.** Altered Slc7a5 sensitivity in S1 mutant Kv1.1 channels.

## Discussion

While the function of core α-subunits of Kv channels has been investigated in depth in the context of voltage-dependent gating, regulation of Kv channels by signaling pathways and regulatory proteins has been less widely studied. Our efforts to identify novel regulatory proteins of Kv channels revealed that Slc7a5, a widely studied amino acid transporter, exerts a powerful influence on gating and expression of Kv1.2 channels (*Baronas et al., 2018*; *Lamothe and Kurata, 2020*). Prominent effects of Slc7a5 include suppression of channel expression, a −50 mV shift of voltage-dependent activation, and marked suppression of current that can be relieved/disinhibited by negative holding potentials. While powerful, the broader significance and underlying molecular mechanisms of these varied effects remain unclear. In this study, we have continued to investigate Slc7a5 regulation of Kv channels to address some of these lingering questions.

We previously demonstrated some Kv1 subtype specificity of Slc7a5 regulation, as Kv1.2 was prominently affected by Slc7a5, while Kv1.5 was not (*Baronas et al., 2018*). Further investigation of other subtypes revealed that Kv1.1 exhibits some of the Slc7a5-mediated hallmark gating features observed in Kv1.2, most notably, the prominent Slc7a5-mediated inhibition relieved by holding the membrane voltage at −120 mV. Surprisingly, this effect persists in the absence of heterologous expression of Slc7a5 (*Figures 2* and *3*). The influence of endogenous levels of Slc7a5 on Kv1.1 was confirmed using an shRNA knockdown approach, along with rescue by shRNA-resistant Slc7a5 cDNA (*Figure 3*). The inhibitory effect of Slc7a5 on Kv1.1 or Kv1.2 is observed immediately upon whole-cell break-in, with current amplitudes significantly reduced compared to expression of either channel alone. It should be noted that after holding at −120 mV, channels typically remain 'disinhibited' such that Kv1 current suppression by Slc7a5 does not re-occur during the recording in most cells. However, other features of Slc7a5 regulation (accelerated inactivation of Kv1.1[Y379T] and Kv1.2[V381T], hyperpolarizing shift of Kv1.2 channel activation) persist even after Slc7a5 inhibition has been relieved. We have not identified a method to accelerate development of the Slc7a5-inhibited state. The underlying mechanism is not yet clear, although the requirement of strong negative voltages (−120 mV) for disinhibition is interesting as it lies outside the voltage range where most gating charge of Kv1.2 channels is displaced (*Goodchild et al., 2012*). At this stage, we have not ruled out whether the strong negative voltages required to relieve Slc7a5-mediated inhibition are related to voltage-dependent conformational changes of the channel, transporter, or both. Interestingly, Kv1.2'LT' (I304L and S308T) mutant channels, with altered coupling between the VSD and the pore, prevent the Slc7a5-mediated inhibition effect but retain sensitivity to the Slc7a5-mediated gating shift in Kv1.2 (*Baronas et al., 2018*). Based on these effects, we hypothesize that Slc7a5-mediated inhibition involves the pore domain (and can also trigger C-type inactivation in channels that are susceptible), which is influenced by coupling to Slc7a5 interactions in the VSD.

We have taken steps to determine whether the Slc7a5-mediated effects on Kv1.1 are due to a direct interaction with the channel, or some indirect effect related to Slc7a5 function. Slc7a5 can indirectly influence mTOR activity (primarily mTORC1), due to activation of mTORC1 by amino acids like leucine (*Nicklin et al., 2009*; *Saxton et al., 2016*; *Wolfson et al., 2016*). Also, mTOR signaling has been shown to modulate Kv1.1 translation in dendrites, although this mechanism is unlikely to be involved here, as it was shown to rely on UTR elements that are not present in our Kv1.1 cDNA (*Niere and Raab-Graham, 2017*; *Raab-Graham et al., 2006*). We used a variety of approaches, including pharmacological inhibition of Slc7a5 (BCH) and mTORC1 (rapamycin) (*Figure 4*), but found no effect on Kv1.1 regulation by Slc7a5. The transport deficient Slc7a5[F252A] mutant also suppressed Kv1.1, although we occasionally observed more prominent recovery of current at −80 mV in Slc7a5[F252A] versus WT Slc7a5 effects in these experiments (*Figure 5—figure supplement 1*, *Figure 6*). Based on recent cryo-EM structures, F252 is located in TM6 of Slc7a5, in very close proximity to the substrate-binding site. Mutations of F252 or other critical sites might influence the conformations achieved by the transporter, and alter its influence on the channel, but this will require further investigation. At present, we have not determined whether there is reciprocal modulation of Slc7a5 function by Kv channels. However, this possibility will continue to be explored, as certain combinations of Kv7 channels and myo-inositol transporters exhibit mutual regulation, along with other channel:transporter combinations (*Abbott et al., 2014*; *Lebowitz et al., 2019*; *Manville et al., 2017*; *Neverisky and Abbott, 2017*).

Slc7a5 is expressed in a wide variety of cell types and undergoes prominent changes in certain physiological or pathological conditions. For example, T cell activation and proliferation requires prominent upregulation of Slc7a5, leading to protein abundance orders of magnitude larger than Kv channel expression (*Marchingo et al., 2020*; *Sinclair et al., 2013*). Slc7a5 upregulation has also been reported in a variety of tumor types (*Barollo et al., 2016*), and Slc7a5 inhibition attenuates proliferation of cells (*Salisbury and Arthur, 2018*). In most instances, the role of Slc7a5 is considered in the context of providing nutrients for cell growth and division, and also activation of mTOR signaling. However, changes in ion channel activity and cellular excitability are also linked to cell division and proliferation in normal development and cancer or other pathologies (*Huang and Jan, 2014*; *Pardo and Stühmer, 2014*; *Satou et al., 2020*; *Serrano-Novillo et al., 2019*; *Yang et al., 2012*). The cellular signals that control ion channel activity and expression during cell growth and division remain unestablished. We will continue to investigate how Slc7a5 or other related proteins may link cellular metabolism with electrical activity, growth, and proliferation. Recent work has also suggested that the inhibitory effect of Slc7a5 on Kv1.1 and Kv1.2 may enhance excitability in neuropathic pain (*Alles et al., 2020*). Gene expression (mousebrain.org) of Slc7a5 overlaps with Kv1.1/1.2 in the NF1-3 subclass of DRG neurons and INH2 subclass of spinal cord neurons, each with varying degrees of expression between cell types. Therefore, excitability in certain cell types in the nervous system may be influenced by Slc7a5-mediated regulation of Kv1 channels, and altered during pain and inflammation. Overall, differential sensitivity, expression, and assembly of Kv1 isoforms and Slc7a5 may fine tune the interplay between channel and transporter. However, physiological modulation of Kv1 channels by Slc7a5 is still under investigation.

Auxiliary subunits can interact with voltage-gated ion channels in a variety of ways. The canonical auxiliary subunits of the Kv1 channel family are the Kvβ subunits, which are soluble proteins that interact with a cytoplasmic scaffolding domain that is structurally distinct from the Kv1 transmembrane domains (*Gulbis et al., 2000*; *Long et al., 2005*). However, there is great diversity amongst known modulators of other Kv channels, including several transmembrane proteins. BK channels can be modulated simultaneously by both the BKβ and BKγ subunits, which are thought to associate with the pore-forming subunits via transmembrane domains (*Gonzalez-Perez et al., 2015*; *Gonzalez-Perez and Lingle, 2019*; *Yan and Aldrich, 2012*, *Yan and Aldrich, 2010*). Similarly, the widely studied KCNE subunits are transmembrane proteins that have been proposed to integrate into clefts between neighboring voltage-sensing domains to modulate channel function (*Murray et al., 2016*; *Wang et al., 2012*; *Xu et al., 2013*). Another Kv channel with prominent regulation of channel gating by multiple classes of regulatory proteins is the Kv4 family, which is sensitive to both KChIP (a soluble cytoplasmic protein) and DPP-like proteins (transmembrane), leading to modulation of channel expression and gating (*Jerng et al., 2005*; *Kitazawa et al., 2015*; *Zagha et al., 2005*). Although we have not yet collected direct evidence of an interaction between Slc7a5 and Kv1.1 or Kv1.2, our findings indicate that Slc7a5 sensitivity is encoded by the voltage-sensing domains of these channels, and is significantly weakened by mutations in the S1 segment. While we suspect this is a site of interaction, Slc7a5 may also influence these channels via an intermediary, or through other nearby regions of the channel that interact allosterically with mutations in S1. Our previous studies have demonstrated co-regulation of Kv1.2 by Slc7a5 and Kvβ subunits (*Baronas et al., 2018*; *Lamothe and Kurata, 2020*), indicating the possibility that Kv1 channels might assemble with multiple accessory subunits simultaneously (similar to the putative Kv4 complex with KChIP and DPP-like proteins). It is also noteworthy that we have previously identified powerful regulation of Kv1.2 gating by extracellular redox conditions, by a mechanism that is not intrinsic to the Kv1.2 α-subunit (*Baronas et al., 2017*). This effect is vastly different from Slc7a5 modulation, indicating an additional regulatory mechanism, and a recent study has suggested a possible role for the sigma opioid receptor (*Abraham et al., 2019*). Thus, there are likely multiple unrecognized regulatory mechanisms that can strongly influence Kv1 channel gating, but it remains unclear how these pathways may interact in cell lines or native tissues.

In summary, our study highlights that Slc7a5 influences multiple targets in the Kv1 family, including Kv1.1 and Kv1.2. Moreover, we map Slc7a5 sensitivity to the voltage-sensing domain of Kv1 channels, highlighting a specific residue in the S1 segment rather than a cytoplasmic signal. We are hopeful that ongoing investigation of Kv channel regulatory proteins will broaden our understanding of voltage-gated channel function and regulation in vivo.

# Materials and methods

**Key resources table**

| Reagent type (species) or resource | Designation | Source or reference | Identifiers | Additional information |
|---|---|---|---|---|
| Cell line (*Mus musculus*) | LM(tk-) cells | ATCC | CCL-1.3 | Fibroblast cells (Male) |
| Cell line (*Homo-sapiens*) | Human Embryonic Kidney 293T (HEK 293T) cells | ATCC | CRL-3216 | |
| Chemical compound, drug | jetPrime | Polyplus | | transfection reagent |
| Gene | GFP | Addgene | 54705 | |
| Gene | mCherry | Addgene | 30125 | |
| Gene (*Rattus norvegicus*) | Kv1.2 | Lab clone | Accession: NM_012970.3 | |
| Gene (*Homo sapiens*) | Kv1.5 | Lab clone | Accession: NM_002234.4 | |
| Gene (*Homo sapiens*) | Kv1.1 | Lab clone | Accession: NM_000217.3 | |
| Gene | eGFP | Addgene | 54759 | |
| Gene (*Homo sapiens*) | Slc7a5 | DNASU | Accession: NM_003486.7 | |
| Gene (*Homo sapiens*) | Slc7a6 | DNASU | Accession: NM_001076785.3 | |
| Recombinant DNA reagent | pcDNA3.1(-) | Invitrogen | | |
| Recombinant DNA reagent | pcDNA3.1(-)-ccdB-Nanoluc | Addgene | 87067 | |
| Strain (*Escherichia-coli*) | DH5-α | Prepared in lab | | |
| Antibody | anti-Slc7a5 (rabbit polyclonal) | Trans Genic Inc | KE026 | 1:1000 dilution |
| Antibody | HRP-conjugated (goat anti-rabbit) | Applied Biological Materials | SH012 | 1:15,000 dilution |
| Antibody | Anti-β actin (mouse monoclonal) | GeneTex | Cat #: GTX629630 | 1:2000 dilution |
| Antibody | HRP-conjugated (goat anti-mouse) | Applied Biological Materials | SH023 | 1:20,000 dilution |
| Antibody | β-tubulin | CST | Ref: #86298, D3U1W | |
| Antibody | Mouse IgG | CST | Ref: #5470 | DyLight 680-conjugated |
| Antibody | Rabbit IgG | CST | Ref: #5151 | DyLight 800-conjugated |
| Antibody | mTOR | CST | Ref: #4517, L27D4 | |
| Antibody | mTOR (phosphor-Ser2448) | CST | Ref: #5536 | |
| Antibody | S6 | CST | Ref: #2317, 54D2 | |
| Antibody | S6 (phosphor-Ser240/244) | CST | Ref: #5364, D68F8 | |
| Antibody | 4-EBP1 | CST | Ref: #9644, 53H11 | |
| Reagent | SuperSignal West Femto Max Sensitivity Substrate | Thermo Fisher Scientific | 34095 | Chemiluminescent detection reagent |
| Chemical compound, drug | 2-Amino-bicyclo[2,2,1] heptane-2-carboxylic acid (BCH) | Sigma Aldrich | A7902-100MG | |

*Continued on next page*

*Continued*

| Reagent type (species) or resource | Designation | Source or reference | Identifiers | Additional information |
|---|---|---|---|---|
| Chemical compound, drug | Rapamycin | Alfa Aesar, Fisher Scientific | J62473-MF | |
| Sequence-based reagent | Slc7a5Taqman probe | Thermo Fisher | Cat #: (Mm00441516_m1) | |
| Sequence-based reagent | GAPDH | Thermo Fisher | Cat #: (Mm99999915_g1) | |
| Commercial assay, kit | Taqman Fast Advance Master Mix | Applied Biosystems | 4444556 | |
| Commercial assay, kit | Superscript IV First-strand synthesis system | Invitrogen | 18091050 | |
| Commercial assay, kit | Nano-Glo live cell assay reagent | Promega | N1620 | |
| Chemical compound, drug | Puromycin | Gibco | Ref #: A11138-03 | |
| Chemical compound, drug | Polybrene | Sigma Aldrich | TR-1003-G | |
| Genetic reagent | pLV-RNAi vector system | Biosettia | Cat #: Sort-B21 | |

## Cell culture and expression

Mouse LM(tk-) fibroblast cells (ATCC CCL-1.3), referred to throughout as LM cells, were used for patch clamp experiments, Western blots, qPCR, and BRET experiments. LM cells were purchased directly from ATCC and screened periodically (3–6 months) for mycoplasma contamination with commercial real time PCR kits. LM(tk-) fibroblasts are not listed on the ICLAC register of commonly misidentified cell lines. Cells were maintained in culture in a 5% $CO_2$ incubator at 37°C in DMEM supplemented with 10% FBS and 1% penicillin/streptomycin. Cells were split into 12-well plates to achieve 70% confluence the subsequent day, then transfected with cDNA using jetPRIME transfection reagent (Polyplus). Fluorescent proteins were co-transfected to identify transfected cells for electrophysiological recording. In 12-well plates, cells were transfected with 400 ng of total DNA plasmid, with the amount of channel DNA plasmid held constant, and either GFP plasmid or mCherry-Slc7a5 (or mEGFP-Slc7a5) in order to maintain a constant amount of DNA in the transfection mixture. For electrophysiological recordings, cells were consistently transfected with 1:1 ratios of channel DNA to eGFP or mCherry-Slc7a5 (unless otherwise indicated). 6–10 hr after transfection, cells were split onto glass coverslips at low density for electrophysiological recordings from single cells the following day. Electrophysiological recordings were done 24–36 hr after transfection.

Stable shRNA-mediated knockdown cell lines were generated by puromycin selection after lentiviral infection of parental LM cells (see 'Lentiviral vector construction and delivery' for details) (Satou et al., 2020). Puromycin-resistant cells were plated in serial dilutions to isolate clonal cell lines. Knockdown cell lines were maintained in the same media as parental LM fibroblasts, along with 2.5 µg/mL puromycin to maintain shRNA expression.

## Potassium channel constructs

Kv1 channel cDNAs (human Kv1.1, rat Kv1.2, human Kv1.5 and various chimeras) were expressed using the pcDNA3.1(-) vector (Invitrogen).

## Kv1.1/Kv1.2 chimeras

Chimeric constructs of human Kv1.1 and rat Kv1.2 were generated using overlapping PCR approaches. N-terminal fragments of Kv1.1 were amplified by PCR using a 5' flanking primer (Kv1.1Forward) and each of the following primers (1.1/1.2-S1; 1.1/1.2-S2; 1.1/1.2-S3; 1.1/1.2-Pore). Additional channel segments from Kv1.2 were amplified using the reverse complement of primers,

and a 3' flanking primer (Kv1.2-Reverse). Primer sequences are listed in an appendix (*Supplementary file 1*).

Resulting full length channel fragments were then cloned into pcDNA3.1(-) using EcoRI and HindIII restriction digests and ligation. This generated chimeric Kv1.1/Kv1.2 ion channels with break points at amino acid numbers, 147, 226, 256, 350 (Kv1.1 numbering, schematics of chimera design are shown in *Figure 5A*). Constructs were all verified by diagnostic restriction digestion and Sanger sequencing (Genewiz, Inc, or University of Alberta Applied Genomics Core).

Chimeric switching of voltage-sensing domains was accomplished using similar overlapping PCR approaches, with break points corresponding to amino acid numbers 147 and 350 (Kv1.1) or 145 and 352 (Kv1.2). To generate the Kv1.2(Kv1.1VSD) chimera we used a previously made Kv1.1/Kv1.2 chimera (breakpoint at Kv1.1 amino acid 350, labeled Kv1.1S6/Kv1.2 in *Figure 6*) and replaced the N terminus with sequence from Kv1.2 using the Kv1.2-Forward and Kv1.2 N-term Reverse primers (see Appendix).

Similarly, the Kv1.1(Kv1.2VSD) chimera was made by switching the pore and C-terminus of our previously made Kv1.1N/Kv1.2 chimera (*Figure 6*) with corresponding Kv1.1 sequence using the Kv1.1 Reverse and Kv1.1 Pore Forward primers (see Appendix, *Supplementary file 1*).

The Kv1.2(1.1-S3/S4) chimera (replacing the S3-S4 linker of Kv1.2 with corresponding sequence from Kv1.1) was generated by amplifying the 3' side of Kv1.2(Kv1.1VSD) with the Kv1.2(1.1-S3/S4)-forward and Kv1.2-Reverse primers, and the 5' side of Kv1.2 with the Kv1.2-Forward and Kv1.2(1.1-S3/S4)-reverse primers. This generated a chimera replacing Kv1.2 residues 274–288 with corresponding sequence from Kv1.1.

## Kv1.2/Kv1.5 chimeras

Chimeras described in *Figure 8—figure supplement 1* were constructed by amplifying segments of Kv1.5 using the Kv1.5-Reverse primer, and each of the following primers S1 chimera; S3 chimera; Pore chimera; C-terminal chimera. These were fused to Kv1.2 sequence using overlapping PCR, and cloned into pcDNA3.1(-) using NheI and HindIII.

Segments of the Kv1.5 S1 and S2 segments were amplified with the following primer sets (1.5S1F and 1.5S1R; 1.5S1S2F and 1.5S1S2R; 1.5S2F and 1.5S2R; 1.5S2S3F and 1.5S2S3R), and used to replace corresponding sequence in Kv1.2 by overlapping PCR. This generated chimeras that replace Kv1.2 sequence with Kv1.5 between amino acid positions (Kv1.2 numbering): 153–187 (S1 chimera); 187–217 (S1-S2L chimera); 217–242 (S2 chimera); 242–269 (S2-S3L chimera).

## Point mutants

Point mutations in Kv1.1 and Kv1.2 were generated with overlapping PCR approaches using the following mutagenic primers and their reverse complement (Kv1.1[Y379T]; Kv1.1[V168I]; Kv1.1[V168A]; Kv1.2[P161S]; Kv1.2[I164A]; Kv1.2[M171L]; Kv1.2[V178I]; Kv1.2[S179T]; Kv1.2[I257F]).

## Electrophysiology

Patch pipettes were manufactured from soda lime capillary glass (Fisher), using a Sutter P-97 puller (Sutter Instrument). When filled with standard recording solutions, pipettes had a tip resistance of 1–3 M$\Omega$. Recordings were filtered at 5 kHz, sampled at 10 kHz, with manual capacitance compensation and series resistance compensation between 70 and 90%, and stored directly on a computer hard drive using Clampex 10 software (Molecular Devices). Bath solution had the following composition: 135 mM NaCl, 5 mM KCl, 1 mM $CaCl_2$, 1 mM $MgCl_2$, 10 mM HEPES, and was adjusted to pH 7.3 with NaOH. Pipette solution had the following composition: 135 mM KCl, 5 mM K-EGTA, 10 mM HEPES and was adjusted to pH 7.2 using KOH. Chemicals for electrophysiological solutions were purchased from Sigma-Aldrich or Fisher. BCH (Slc7a5 inhibitor 2-amino-bicyclo[2,2,1]heptane-2-carboxylic acid, Sigma-Aldrich) was stored as a 100 mM stock solution in 1N NaOH, and diluted to working concentrations each experimental day. Rapamycin (Alfa Aesar, Fisher Scientific) was stored as a 10 mM stock solution in DMSO and diluted to working concentrations each experimental day.

## Lentiviral vector construction and delivery

We generated shRNAs targeting four segments of mouse Slc7a5 (refseq: NM_011404), with the ShR1, ShR2, ShR3, ShR4, and Negative control primers (see Appendix). Oligos were designed for

hairpin formation and cloned into pLV-RNAi vector system (Biosettia, San Diego, USA. Cat#sort-B21) according to the manufacturer's instructions, as previously described (*Satou et al., 2020*). All constructs were confirmed by Sanger sequencing (Applied Genomics Core, University of Alberta). HEK293T cells were co-transfected with packaging vectors (Provided in the pLV-RNAi vector system) and an shRNA expression vector. Lentivirus was harvested by centrifugation of HEK293T cell culture supernatant 48 hr after transfection. LM fibroblasts cells were seeded in six well plate at about 30% confluence the day before viral transduction. After 24 hr, media was replaced with fresh complete medium (DMEM with 10% FBS) containing 8 µg/mL polybrene and 0.5 mL of viral supernatant. After 24 hr of incubation, the virus-containing medium was replaced with fresh complete medium. After further incubation for 48 hr, puromycin (2.5 µg/mL) was added and maintained in culture for selection of transduced cells. For the generation of clonal shRNA-expressing cell lines, transduced cells were plated by serial dilution in 96 well plates, and wells with single cells were identified by visual inspection under a microscope. After expansion of individual clones, effectiveness of knockdown was assessed using Western blot and qPCR approaches.

## Western blot

### Detection of Slc7a5

Cell lysates from LM fibroblasts were harvested in NP-40 lysis buffer (1% NP-40, 150 mM NaCl, 50 mM Tris-HCl) with 1% protease inhibitor cocktail (Sigma, P8340), 3 days after transfection. Samples were separated on 8% SDS-PAGE gels, and transferred to nitrocellulose membranes using standard Western blot apparatus (Bio-rad). Running buffer composition was 190 mM Glycine, 25 mM Tris, 4 mM SDS. Transfer buffer composition was 20% methanol, 3.5 mM $Na_2CO_3$, 10 mM $NaHCO_3$. Slc7a5 was detected using a rabbit polyclonal Slc7a5 antibody (1:500 dilution, KE026; Trans Genic Inc) and HRP-conjugated goat anti-rabbit antibody (1:15 000 dilution, SH012; Applied Biological Materials). Chemiluminescence was detected using SuperSignal West Femto Max Sensitivity Substrate (Thermo Fisher Scientific) and a FluorChem SP gel imager (Alpha Innotech).

### Detection of phosphorylated and total mTOR, S6 and 4-EBP1

LM fibroblasts cells (parental LM or ShR4-1) were lysed for 2 hr at 4°C in NP-40 lysis buffer (1% NP-40, 150 mM NaCl, 50 mM Tris-HCl) with 1% protease inhibitor cocktail (Sigma, P8340) and 1% phosphatase inhibitor (Sigma, P5726). 18 µg of proteins were separated using 4–20% Mini-PROTEAN precast gel (BioRad). Subsequently, these proteins were transferred onto PVDF membranes (Millipore). The membrane was blocked by 5% skim milk dissolved with a wash buffer (Tris/HCl pH 7.5, 0.05% Tween-20) for 1 hr at room temperature. The membrane was incubated with the appropriate primary antibody (listed in the corresponding table) followed by the compatible Fluorescein-conjugated secondary antibody (listed in the corresponding table). Fluorescence was detected by the Odyssey Imaging System (LI-COR Biosciences).

| Target | Merchandiser | Remarks |
| --- | --- | --- |
| β-tubulin | CST | #86298, D3U1W |
| Mouse IgG | CST | #5470, DyLight 680-conjugated |
| Rabbit IgG | CST | #5151, DyLight 800-conjugated |
| mTOR | CST | #4517, L27D4 |
| mTOR (phosho-Ser2448) | CST | #5536 |
| S6 | CST | #2317, 54D2 |
| S6 (phospho-Ser240/244) | CST | #5364, D68F8 |
| 4-EBP1 | CST | #9644, 53H11 |

## Quantification of Slc7a5 mRNA expression using qPCR

Total cellular RNA was extracted from LM fibroblasts stably expressing shRNA constructs using an illustra RNAspin Mini kit (GE, UK). RNA concentration was assessed using a Nanodrop 2000c spectrophotometer (Thermo Scientific). Reverse transcription was performed using the SuperScript IV

First-Strand Synthesis System (Invitrogen, USA), using Oligo(dT)20 primers to make cDNA. Real-time quantitative PCR was carried out with TaqMan Fast Advance Master Mix (Applied Biosystems, USA) in an ABI 7900HT Fast Real Time PCR System (Applied Biosystems). Taqman probes used were: Slc7a5 (cat: Mm00441516_m1), and GAPDH as an internal control gene (cat: Mm99999915_g1), obtained from Thermo Fisher. The cycling protocol used an initial denaturation at 95℃ for 3 min; 40 cycles of denaturation at 95℃ for 3 s, annealing at 60℃ for 30 s. Data were analyzed using the $2^{-\Delta\Delta CT}$ method (*Livak and Schmittgen, 2001*) and expressed as Slc7a5 normalized to GAPDH.

## Bioluminescence resonance energy transfer

Nanoluc was amplified from pcDNA3.1-ccdB-Nanoluc (a gift from Mikko Taipale, Addgene # 87067), and fused to the C-terminus of Kv1.1 in pcDNA3.1(-) using EcoRI and HindIII restriction sites. Using standard subcloning methods, WT Kv1.1, Slc7a5 and Slc7a6 cDNAs were tagged at the N-terminus with mEGFP. Mouse LM fibroblast cells were plated at 30% confluency in 12-well plates and then transiently transfected using jetPRIME the next day with cDNAs encoding BRET donors and acceptors. 48 hr after transfection, cells were replated onto white polystyrene 96-well plates (Thermo Fisher). After 24 hr, cells were washed with PBS, and incubated with Nano-Glo live cell assay reagent (Promega). Emission spectra were measured between 400 and 700 nm in 2 or 5 nm increments, and measured for 2 s at each interval with a Synergy H4 Hybrid Reader (BioTek). All Spectral data were normalized to the peak nanoluc emission, and subtraction of the normalized Kv1.1-nanoluc spectrum (measured as a blank control) produced the EGFP emission. The area under the curve (AUC) for the integrated EGFP emissions of Slc7a5 and Slc7a6 were normalized to the integrated EGFP emission from Kv1.1 nanoluc + EGFP-Kv1.1 in each experiment.

## Data analysis

Wherever possible, we have displayed data for all individual data points collected, in addition to reporting mean ±S.D. or a box plot (where shown, box plots depict the median, 25th and 75th percentile (box), and 10th and 90th percentile (whiskers)). Conductance-voltage relationships were fit with a Boltzmann equation (*Equation 1*), where G is the normalized conductance, V is the applied voltage, $V_{1/2}$ is the half activation voltage, and $k$ is a fitted slope factor reflecting the steepness of the curve.

$$G = 1/(1 + e^{-(v - v_{1/2})/k}) \tag{1}$$

Conductance-voltage relationships were fit for each individual cell, and the extracted fit parameters were used for statistical calculations. Statistical tests are described in corresponding figure legends throughout the manuscript.

Data files containing numerical data for each figure and their supplements have been associated with the article, along with a list of primer sequences (*Figure 1—source data 1*, *Figure 2—source data 1*, *Figure 3—source data 1*, *Figure 4—source data 1*, *Figure 5—source data 1*, *Figure 6— source data 1*, *Figure 7—source data 1*, *Figure 8—source data 1*, *Figure 9—source data 1*, *Supplementary file 1*).

## Acknowledgements

This work was funded by a Canadian Institutes of Health Research Project Grant to HTK. SML was supported by a Dr. Rowland and Muriel Haryett Neuroscience Fellowship, University of Alberta Neuroscience and Mental Health Institute. VAB was supported by a Canadian Institutes of Health Research Vanier award. GS was supported by a Natural Sciences and Engineering Research Council USRA award. During early stages of this project, HTK was supported by a Canadian Institutes of Health Research Early Career Investigator award and salary support from the Alberta Diabetes Institute.

## Additional information

### Funding

| Funder | Grant reference number | Author |
|---|---|---|
| Canadian Institutes of Health Research | Project Grant | Harley T Kurata |
| Canadian Institutes of Health Research | Vanier Studentship | Victoria A Baronas |
| University of Alberta | Rowland and Muriel Haryett fellowship | Shawn M Lamothe |
| Natural Sciences and Engineering Research Council of Canada | USRA | Grace Silver |
| Canadian Institutes of Health Research | Early Career Investigator | Harley T Kurata |
| Alberta Diabetes Institute | Salary support | Harley T Kurata |

The funders had no role in study design, data collection and interpretation, or the decision to submit the work for publication.

### Author contributions

Shawn M Lamothe, Conceptualization, Formal analysis, Funding acquisition, Investigation, Writing - review and editing; Nazlee Sharmin, Victoria A Baronas, Conceptualization, Investigation, Writing - review and editing; Grace Silver, Investigation, Writing - review and editing; Motoyasu Satou, Resources, Investigation, Methodology, Writing - review and editing; Yubin Hao, Investigation, Methodology, Writing - review and editing; Toru Tateno, Resources, Supervision, Investigation, Methodology, Writing - review and editing; Harley T Kurata, Conceptualization, Data curation, Formal analysis, Supervision, Funding acquisition, Investigation, Visualization, Methodology, Writing - original draft, Project administration, Writing - review and editing

### Author ORCIDs

Shawn M Lamothe (iD) https://orcid.org/0000-0001-8722-2631
Harley T Kurata (iD) https://orcid.org/0000-0003-4357-4189

### Decision letter and Author response

Decision letter https://doi.org/10.7554/eLife.54916.sa1
Author response https://doi.org/10.7554/eLife.54916.sa2

## Additional files

### Supplementary files
- Supplementary file 1. List of primers.
- Transparent reporting form

### Data availability

All data generated or analysed during this study are included in the manuscript and supporting files. Source data files have been provided for all figures.

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
