## [Decision Letter]

**Acceptance summary:**

Recent studies have shown that the activity of some families of voltage-sensing potassium channel is modulated by transporters, suggesting that they interact in hitherto unanticipated manner. This study by Lamothe et al. shows that the amino acid transporter Slc7a5 broadly modulates the activity of other potassium channel isoforms but the functional outcomes are different. These findings highlight the diversity of regulatory mechanisms and enable identification of crucial regions involved in mediating these novel interactions.

**Decision letter after peer review:**

Thank you for submitting your work entitled "Control of Slc7a5 sensitivity by the voltage-sensing domain of Kv1 channels" for consideration by *eLife*. Your article has been reviewed by three peer reviewers, including Baron Chanda as the Reviewing Editor and Reviewer #1, and the evaluation has been overseen by a Senior Editor. The following individual involved in review of your submission has agreed to reveal their identity: Geoff Abbott (Reviewer #2).

Our decision has been reached after consultation between the reviewers. Based on these discussions and the individual reviews below, we regret to inform you that your work will not be considered further for publication in *eLife*.

The study identifies a new target for regulation by SLC75a5 which has been previously shown to regulate Kv1.2. This study shows that this regulation extends to other Kv1 channels but in ways that are unexpectedly different. These studies provide new insights into a relatively new area of ion channel biology. From a mechanistic standpoint, the study also shows that the voltage-sensing domain is the key domain involved in mediating these disparate effects on channel function. Nonetheless, the consensus amongst reviewers is that the manuscript does not provide sufficient mechanistic insight to warrant further consideration in *eLife*. As you will see below, there are some differences in opinion as to what will be sufficient. I hope the reviewers' feedback will help you improve the manuscript. If you believe you will be able to address these concerns and would like to reach out to us with a planned resubmission, please do not hesitate to contact us in advance.

Reviewer #1:

The manuscript "Control of Slc7a5 sensitivity by the voltage sensing domain of Kv1 channels" by Sharmin et al., is an extension of the previous work from the same lab showing the effects of Slc7a5 co-expression on gating and expression of Kv1.2. In this manuscript the authors show that Slc7a5 can also regulate Kv1.1 with slightly different effects. Kv1.1 appears to be modulated by Slc7a5 even at the endogenous levels of expression. Furthermore, swapping the voltage sensing domains between Kv1.1 and Kv1.2 can also swap Slc7a5 sensitivity. The data presentation is transparent and the manuscript is well-written. However, this work would have been much more impactful if the authors had gone ahead and tried to identify the molecular mechanisms that contribute to differential sensitivity instead of limiting themselves to low-resolution domain level view. Given our understanding of the structure of Kv1 channel and the homology between the two isoforms, it seems this would be rather straightforward.

Some of the concerns I have regarding the manuscript are as follows:

1) Considering that kv1.1 is susceptible to endogenous levels of Slc7a5, could it be possible that the G-V curves shown in Figure 1 are already left shifted? Have the authors recorded the G-V curves from the endogenous Slc7a5 knock out cell line (shR41 used in Figure 4D and 4E)? If yes, do they see a shift in V1/2 when they rescue the cell line with human Slc7a5?

2) The authors should include in the discussion what could be the putative physiological importance of such differential sensitivity of two isoforms of Kv1. Along the same lines, have the authors tested the effects of Slc7a5 co-expression on the concatenated heterodimers of Kv1.1 and Kv1.2 that the authors have used in one of the previous manuscripts. This could be important considering the physiological existence of these isoforms potentially as heterodimers.

3) Do authors believe that 6 hours of rapamycin treatment would completely inhibit downstream mTOR signaling? Although mTORC1 activity is inhibited completely within 30 minutes of rapamycin exposure, mTORC2 inhibition would require longer incubation than 6 hours (multiple lines of evidence, for example Sarbassov et al., 2006). Have the authors done any experiments to show that the concentration used in the manuscript for 6 hours can inhibit mTOR signaling completely? If not, these details need to be mentioned in the discussion.

Reviewer #2:

The authors describe regulation of the Kv1.1 potassium channel by the sodium-coupled amino acid transporter, Slc7a5. This is a follow-up to their previous publication showing Slc7a5 regulation of Kv1.2. The main difference between the two is that Slc7a5 reduces Kv1.2 current density and negatively shifts its voltage dependence of activation; Slc7a5 does not alter Kv1.1 in this way but instead alters disinhibition from a presumed non-conducting state and speeds C-type inactivation of the hyper-inactivating Kv1.1-Y379T mutant. The voltage sensor appears to dictate which of these sets of effects occur, based on convincing chimera data. Overall this is interesting work that pushes forward the relatively new field of channel-transporter complexes.

Specific points.

1) Have the authors considered that rather than disinhibiting Kv1.1 and Kv1.2 from a non-conducting state, Slc7a5 has the opposite effect, and slows disinhibition from this state? In many of the examples they show, the channels reach an equivalent or greater peak current after the -120 mV pulses in the absence of Slc7a5 (either immediately or after a few pulses) versus in the presence. When Slc7a5 is co-expressed, the channel eventually achieves a similar current level, but it takes many more pulses. It appears more as if the transporter slows disinhibition of the channel and/or favors the equilibrium toward the inhibited state. The authors should demonstrate that they can distinguish between the two completely opposite conclusions.

2) Related to the point above, if the non-conducting state is actually a form of C-type inactivation, then slowed disinhibition (recovery) from inactivation with Slc7a5 would be consistent with the more rapid inactivation they observe for the mutant Kv1.1 channel when Slc7a5 is also present. The authors should try some inactivation recovery voltage protocols in which they vary the recovery pulse voltage and duration as this might provide answers or at least clues to points 1 and 2.

3) Still on the inactivation theme – does Slc7a5 alter the slower C-type inactivation that wild-type Kv1.1 undergoes with longer pulses? This is more important than the unnatural process shown with the Y379T mutant.

4) Some evidence of physical interaction would strengthen the manuscript.

Reviewer #3:

This research group had reported the regulation of Kv1.2 by Slc7a5. This manuscript reports the difference in the effects of Slc7a5 on the gating of Kv1.2 and Kv1.1 and found that the voltage sensor domain underlies the difference. However, this new study provides only some limited incremental insight on the mechanisms of Kv1 channel regulation by Slc7a5. The finding that the voltage sensor domain underlies the Kv1.1 and kv1.2 difference is new but without surprise. Importantly, the mechanisms on how Slc7a5 affects Kv channel gating via its impact on voltage sensor domain remains completely unknown.

1) Results section. The difference in disinhibition between Kv1.1 and Kv1.2 caused by Slc7a5 could be explained by the different effects on their inactivation. It is likely that Kv1.2 shows no inactivation in the absence of exogenous Slc7a5 co-expression while Kv1.1 undergoes partial inactivation even with endogenous Slc7a5 expression. Both Kv1.2 and Kv1.1 could be similarly inactivated with exogenous Slc7a5 expression. A comparison in inactivation rates and levels (i.e., portion of channels became inactivated) between Kv1.1 and Kv1.2 with endogenous Slc7a5 only and in the presence of exogenous Slc7a5 should be helpful to explain the difference in disinhibition which is essentially a recovery from the inactivated state at very negative membrane voltages.

2) Subsection “Inhibition of Slc7a5 transport function or signaling does not prevent Kv1 modulation”. Although inhibition of the Slc7a5 transporter activity and the downstream mTOR activity had little effect on Kv1.1 channel's sensitivity to Slc7a5, it remains unclear whether there are direct physical interactions involved. Particularly, how voltage sensor domain is involved in the Slc7a5's effects on Kv channel gating is completely unknown. It will be helpful if some biochemical experiments can be done to demonstrate physical interactions between Slc7a5 and kv1.1 or 1.2 voltage sensor domain. In addition, the amino acid sequences of voltage sensor domains between Kv1.1 and 1.2 are highly similar, it won't be difficult to identify amino acid residues involved for the 2 channels' difference in their sensitivities to Slc7a5.

[Editors’ note: further revisions were suggested prior to acceptance, as described below.]

Thank you for resubmitting your article "Control of Slc7a5 sensitivity by the voltage-sensing domain of Kv1 channels" for consideration by *eLife*. Your article has been reviewed by three peer reviewers, including Baron Chanda as the Reviewing Editor and Reviewer #1, and the evaluation has been overseen by Kenton Swartz as the Senior Editor. The following individuals involved in review of your submission have agreed to reveal their identity: Geoff Abbott (Reviewer #2).

The reviewers have discussed the reviews with one another and the Reviewing Editor has drafted this decision to help you prepare a revised submission.

Summary:

The manuscript "Control of Slc7a5 sensitivity by the voltage sensing domain of Kv1 channels" by Sharmin et al., is an extension of the previous work from the same lab showing the effects of Slc7a5 co-expression on gating and expression of Kv1.2. In this manuscript the authors show that Slc7a5 can also regulate Kv1.1 with slightly different effects. Kv1.1 appears to be modulated by Slc7a5 even at the endogenous levels of expression. Furthermore, swapping the voltage sensing domains between Kv1.1 and Kv1.2 can also swap Slc7a5 sensitivity. In the revised version, new data to differentiate between Slc7a5-induced inhibition and classic C-type inactivation was added. Furthermore, they have able to identify the residues that are responsible for the difference in Slc7a5's modulation of Kv1 channels' gating properties. The data clearly demonstrate that the voltage sensor domain (VSD) is relevant in the different effects of Slc7a5 on Kv1.1, 1.2 and 1.5 channels.

Revisions:

Some concerns remain.

1) It is helpful to provide some characterization or explanation of the Slc7a5-induced Kv1.1 and 1.2 channel inhibition which can be disinhibited by holding at -120 mV. How does the inhibition occur? It is not shown in the data what condition and how long needed for the inhibition to occur. After disinhibition by holding at a very negative voltage, it is unclear whether the channel can go back to the inhibited state.

2) It is possible that Slc7a5 may act on VSDs either directly via physical interactions or indirectly via another protein. However, the current data cannot rule out other possibilities, e.g., Slc7a5 may act on other parts of the channels which somehow allosterically magnifies or modifies the Kv1 channels' intrinsic voltage gating properties.

3) It is interesting that the F252A mutation on Slc7a5 exerted some effect on Kv1 gating. The observed interaction between inactivation and inhibition is also interesting. Some elaboration or thoughts on those could be helpful for readers.

---

## [Author Response]

[Editors’ note: The authors appealed the original decision. What follows is the authors’ response to the first round of review.]

Reviewer #1:The manuscript "Control of Slc7a5 sensitivity by the voltage sensing domain of Kv1 channels" by Sharmin et al., is an extension of the previous work from the same lab showing the effects of Slc7a5 co-expression on gating and expression of Kv1.2. In this manuscript the authors show that Slc7a5 can also regulate Kv1.1 with slightly different effects. Kv1.1 appears to be modulated by Slc7a5 even at the endogenous levels of expression. Furthermore, swapping the voltage sensing domains between Kv1.1 and Kv1.2 can also swap Slc7a5 sensitivity. The data presentation is transparent and the manuscript is well-written. However, this work would have been much more impactful if the authors had gone ahead and tried to identify the molecular mechanisms that contribute to differential sensitivity instead of limiting themselves to low-resolution domain level view. Given our understanding of the structure of Kv1 channel and the homology between the two isoforms, it seems this would be rather straightforward.Some of the concerns I have regarding the manuscript are as follows:1) Considering that kv1.1 is susceptible to endogenous levels of Slc7a5, could it be possible that the G-V curves shown in Figure 1 are already left shifted? Have the authors recorded the G-V curves from the endogenous Slc7a5 knock out cell line (shR41 used in Figure 4D and 4E)? If yes, do they see a shift in V1/2 when they rescue the cell line with human Slc7a5?

This experiment is now included in the revised manuscript (Figure 3—figure supplement 2). The Kv1.1 voltage-dependence of activation is not affected by Slc7a5 overexpression or knockout in the shR4-1 cell line. This modulation of voltage-dependent activation is a clear difference between Kv1.2 and Kv1.1.

2) The authors should include in the discussion what could be the putative physiological importance of such differential sensitivity of two isoforms of Kv1. Along the same lines, have the authors tested the effects of Slc7a5 co-expression on the concatenated heterodimers of Kv1.1 and Kv1.2 that the authors have used in one of the previous manuscripts. This could be important considering the physiological existence of these isoforms potentially as heterodimers.

We have added a discussion of physiological systems where Slc7a5 modulation of Kv1 channels may arise (Discussion section). A few key points are that Slc7a5 is markedly upregulated in conditions of rapid cell growth and division (ie. tumors, T-cell activation and expansion), and may influence Kv1 channel function. A recent publication has also hypothesized a role for Slc7a5 in regulating neuronal excitability in pain and inflammation (Alles et al., 2020). The relative abundance of Kv1 subtypes with different Slc7a5 sensitivity may influence the outcome in these scenarios. These are a few potential scenarios, although a clear cut physiological system illustrating Slc7a5 modulation of Kv1 channels has not been established.

As requested we have added data from the Kv1.2-Kv1.1 tandem heterodimer described in: Baronas et al., 2015 in Figure 3—figure supplement 3. These heterodimeric channels exhibit shifted voltage-dependence of activation (but not as large as seen for Kv1.2 homomers), and attenuated Slc7a5-mediated inhibition compared to Kv1.1 homomers. We speculate that this difference is likely due to the stoichiometric requirements for these two functional outcomes.

3) Do authors believe that 6 hours of rapamycin treatment would completely inhibit downstream mTOR signaling? Although mTORC1 activity is inhibited completely within 30 minutes of rapamycin exposure, mTORC2 inhibition would require longer incubation than 6 hours (multiple lines of evidence, for example Sarbassov et al., 2006). Have the authors done any experiments to show that the concentration used in the manuscript for 6 hours can inhibit mTOR signaling completely? If not, these details need to be mentioned in the discussion.

Thank you for bringing this up, we have clarified this in the revised manuscript. Our primary aim was to investigate the possibility of a signal-mediated effect of Slc7a5 on Kv1.1. There is some consensus that mTORC1 is the primary amino acid responsive mTOR complex, so this was our primary concern (Saxton and Sabatini, 2017). We have expanded significantly on these considerations (together with some other reviewer suggestions) in the paper as follows:

In addition to rapamycin, we have now included data from a transport deficient Slc7a5[F252A] mutant, along with BCH (Slc7a5 transport inhibitor), which demonstrate that the transport function of Slc7a5 is not required for Kv1.1 modulation. These conditions recapitulate the gating effects of WT Slc7a5 on Kv1.1, suggesting that signals related to the transport function of Slc7a5 (amino acid uptake) are not essential for Kv1.1 modulation (see Figures 4, 5, 6 and accompanying supplements).

We also included western blot data demonstrating basal levels of mTORC1 activation based on phospho-S6, phospho-mTOR and 4-EBP1 detection, along with complete inhibition by rapamycin (Figure 4, and Figure 4—figure supplement 1).

We have also stated explicitly that the experiments do not absolutely rule out the involvement of mTORC2, however mTORC1 is the primary amino acid responsive mTOR complex.

Reviewer #2:The authors describe regulation of the Kv1.1 potassium channel by the sodium-coupled amino acid transporter, Slc7a5. This is a follow-up to their previous publication showing Slc7a5 regulation of Kv1.2. The main difference between the two is that Slc7a5 reduces Kv1.2 current density and negatively shifts its voltage dependence of activation; Slc7a5 does not alter Kv1.1 in this way but instead alters disinhibition from a presumed non-conducting state and speeds C-type inactivation of the hyper-inactivating Kv1.1-Y379T mutant. The voltage sensor appears to dictate which of these sets of effects occur, based on convincing chimera data. Overall this is interesting work that pushes forward the relatively new field of channel-transporter complexes.Specific points.1) Have the authors considered that rather than disinhibiting Kv1.1 and Kv1.2 from a non-conducting state, Slc7a5 has the opposite effect, and slows disinhibition from this state? In many of the examples they show, the channels reach an equivalent or greater peak current after the -120 mV pulses in the absence of Slc7a5 (either immediately or after a few pulses) versus in the presence. When Slc7a5 is co-expressed, the channel eventually achieves a similar current level, but it takes many more pulses. It appears more as if the transporter slows disinhibition of the channel and/or favors the equilibrium toward the inhibited state. The authors should demonstrate that they can distinguish between the two completely opposite conclusions.

Thank you for requesting this clarification – we regret that we did not describe this aspect of our interpretation more clearly in the initial submission. We intended to convey that Slc7a5 has an inhibitory effect on Kv1.1 and Kv1.2 (causing very small initial currents after whole cell break in). This effect is then relieved by very negative holding voltages – this relief of inhibition is what we had referred to as “disinhibition”. Addition of more Slc7a5 makes this effect more prominent, (ie. Figure 2B, E, F, Figure 3E-G). This may reflect a higher stoichiometry of interaction of the channel with overexpressed Slc7a5 (i.e. occupancy of more subunits than with lower levels of endogenous Slc7a5).

We have attempted to clarify this throughout the text. For example, we switched to primarily referring to “relief of Slc7a5-mediated inhibition”. We feel this will help and is more descriptive of our observations. We have also more clearly described the effect and its voltage-dependence in our discussion of Figure 2, 3 and 6.

2) Related to the point above, if the non-conducting state is actually a form of C-type inactivation, then slowed disinhibition (recovery) from inactivation with Slc7a5 would be consistent with the more rapid inactivation they observe for the mutant Kv1.1 channel when Slc7a5 is also present. The authors should try some inactivation recovery voltage protocols in which they vary the recovery pulse voltage and duration as this might provide answers or at least clues to points 1 and 2.

This is an interesting aspect of this system. We have added significantly more detail, although we must admit that we do not yet understand completely. This information has been primarily added to Figures 5 and 6.

A clear outcome of our findings is that Slc7a5-mediated inhibition is distinct from C-type inactivation. In Figure 6 (and Figure 6—figure supplement 1), we demonstrate that relief of Slc7a5-mediated inhibition requires a strong hyperpolarization to -120 mV (hyperpolarization to -80 mV is not effective). In contrast, after channels have been rescued from Slc7a5 inhibition, they can undergo C-type inactivation, and recover readily from C-type inactivation with a holding voltage of -80 mV. This is shown in Figure 6, and more dramatically for the C-type enhanced Y379T mutant in Figure 6—figure supplement 1. This illustrates that Slc7a5-mediated inhibition is distinct from C-type inactivation, although the effects of Slc7a5 on in Kv1.1[Y379T] and Kv1.2[V381T] mutants suggest there may be some interplay between these processes.

3) Still on the inactivation theme – does Slc7a5 alter the slower C-type inactivation that wild-type Kv1.1 undergoes with longer pulses? This is more important than the unnatural process shown with the Y379T mutant.

We have reported the effects of Slc7a5 on inactivation of WT Kv1.1 in the revised manuscript. We do not observe a consistent change in the time course of C-type inactivation in WT Kv1.1, even though these channels exhibit strong inhibition by Slc7a5. This finding further supports our overall suggestion that Slc7a5-mediated inhibition is distinct from C-type inactivation. This is shown in Figure 5 of the revision (previous experiments on Kv1.1[Y379T] have been moved to Figure 5—figure supplement 1 and updated to include data from the Slc7a5[F252A] transport-deficient mutant).

4) Some evidence of physical interaction would strengthen the manuscript.

We agree and we have tried various approaches for co-immunoprecipitation with Slc7a5, but we have not been able to demonstrate this convincingly. In the revised manuscript, we have used a BRET approach to demonstrate proximity of Kv1.1 and Slc7a5 (similar to what we reported for Kv1.2 and Slc7a5 in Baronas et al., 2018). We have included this in the revision in Figure 4, although we have qualified our findings and clarified that this is not necessarily a demonstration of direct association.

We have also added considerable chimeric/mutational analysis of Slc7a5 sensitivity in this revision. Additional data in Figures 8 and 9 identify a particularly important position in S1 (Kv1.2 I164/ Kv1.1 V168). Again this does not demonstrate physical association but begins to reveal specific structural regions that influence susceptibility to Slc7a5-mediated modulation.

Reviewer #3:This research group had reported the regulation of Kv1.2 by Slc7a5. This manuscript reports the difference in the effects of Slc7a5 on the gating of Kv1.2 and Kv1.1 and found that the voltage sensor domain underlies the difference. However, this new study provides only some limited incremental insight on the mechanisms of Kv1 channel regulation by Slc7a5. The finding that the voltage sensor domain underlies the Kv1.1 and kv1.2 difference is new but without surprise. Importantly, the mechanisms on how Slc7a5 affects Kv channel gating via its impact on voltage sensor domain remains completely unknown.1) Results section. The difference in disinhibition between Kv1.1 and Kv1.2 caused by Slc7a5 could be explained by the different effects on their inactivation. It is likely that Kv1.2 shows no inactivation in the absence of exogenous Slc7a5 co-expression while Kv1.1 undergoes partial inactivation even with endogenous Slc7a5 expression. Both Kv1.2 and Kv1.1 could be similarly inactivated with exogenous Slc7a5 expression. A comparison in inactivation rates and levels (i.e., portion of channels became inactivated) between Kv1.1 and Kv1.2 with endogenous Slc7a5 only and in the presence of exogenous Slc7a5 should be helpful to explain the difference in disinhibition which is essentially a recovery from the inactivated state at very negative membrane voltages.

We agree this is an important question. As mentioned above, we have added experiments and clarified our writing to describe how Slc7a5-mediated inhibition is distinct from C-type inactivation. We report that WT Kv1.1 is clearly susceptible to Slc7a5-mediated inhibition, although there is not a consistent effect on inactivation (see Figure 5). We also demonstrate that relief of Slc7a5 inhibition vs. C-type inactivation occurs at very different holding voltages (see Figure 6, and Figure 6—figure supplement 1). Notably, even after nearly complete inactivation of Kv1.1[Y379T] channels (Figure 6—figure supplement 1), they recover readily with a holding voltage of -80 mV. This suggests that Slc7a5-mediated inhibition is distinct from C-type inactivation.

2) Subsection “Inhibition of Slc7a5 transport function or signaling does not prevent Kv1 modulation”. Although inhibition of the Slc7a5 transporter activity and the downstream mTOR activity had little effect on Kv1.1 channel's sensitivity to Slc7a5, it remains unclear whether there are direct physical interactions involved. Particularly, how voltage sensor domain is involved in the Slc7a5's effects on Kv channel gating is completely unknown. It will be helpful if some biochemical experiments can be done to demonstrate physical interactions between Slc7a5 and kv1.1 or 1.2 voltage sensor domain. In addition, the amino acid sequences of voltage sensor domains between Kv1.1 and 1.2 are highly similar, it won't be difficult to identify amino acid residues involved for the 2 channels' difference in their sensitivities to Slc7a5.

We agree these are important concerns and we have added additional data to address these questions, particularly in Figures 4, 7, 8 and 9 of the revised manuscript. We had initially pursued more detailed experiments of Kv1.1/1.2 chimeras. These data are now included as supplements to Figure 7, although we found these experiments to be ambiguous in that we could not isolate individual voltage sensor regions that determined a Kv1.1- vs. Kv1.2-like response. This may be because both Kv1.1 and Kv1.2 are sensitive to Slc7a5, leading to mixed effects on the gating shift and Slc7a5-mediated inhibition in these chimers. Although We could not find a clear answer using the approach of mixing segments of Kv1.1 and Kv1.2, we agree that this may be useful information and is worthwhile to include.

It turned out that a far more useful approach was to use chimeric channels comprising segments of Kv1.5 (which is insensitive to any Slc7a5 modulation, as far as we can tell). This approach was much more fruitful, and revealed a single amino acid in the S1 segment that strongly influences Slc7a5 sensitivity. This has been included in Figures 8 and 9 of the revision, highlighting that residue Kv1.2[164] or Kv1.1[168] strongly influences the response to Slc7a5.

As mentioned to reviewer #2, we have not successfully co-immunoprecipitated Kv1.1 and Slc7a5 (or Kv1.2 and Slc7a5), which may be a reflection of the interaction being detergent sensitive, or implying the involvement of additional proteins. We have used a BRET assay that illustrates proximity in the revised paper (Figure 4D-F), although we recognize that more work will need to be done to determine the structural basis for modulation. We have tried to convey this uncertainty regarding a physical association clearly in the revision.

[Editors’ note: what follows is the authors’ response to the second round of review.]

Revisions:Some concerns remain.1) It is helpful to provide some characterization or explanation of the Slc7a5-induced Kv1.1 and 1.2 channel inhibition which can be disinhibited by holding at -120 mV. How does the inhibition occur? It is not shown in the data what condition and how long needed for the inhibition to occur. After disinhibition by holding at a very negative voltage, it is unclear whether the channel can go back to the inhibited state.

We have now addressed this in the Results section, along with more details in the Discussion (see also point 3). We clarify that Slc7a5-mediated inhibition is observed immediately after break-in. After relief of inhibition, currents typically remain stable for the remainder of the experiment. We presume that Slc7a5-mediated inhibition occurs over longer time scales than we are able to observe in a patch clamp recording.

2) It is possible that Slc7a5 may act on VSDs either directly via physical interactions or indirectly via another protein. However, the current data cannot rule out other possibilities, e.g., Slc7a5 may act on other parts of the channels which somehow allosterically magnifies or modifies the Kv1 channels' intrinsic voltage gating properties.

We agree that although we have identified specific residues that influence the functional effects of Slc7a5 on Kv channels, this does not demonstrate that this is the specific site of interaction. We have made a note of this towards the end of the Discussion section.

3) It is interesting that the F252A mutation on Slc7a5 exerted some effect on Kv1 gating. The observed interaction between inactivation and inhibition is also interesting. Some elaboration or thoughts on those could be helpful for readers.

We agree that the channel inhibition by the F252A mutant is an interesting finding because it suggests that modulation is not dependent on signaling pathways typically related to Slc7a5. Based on our findings we favor the explanation of a direct association between the channel and transporter, although this will require deeper investigation to confirm. We have added a few additional thoughts on this topic in the Discussion. In terms of the relationship between inactivation and Slc7a5 mediated inhibition, we have alluded to this in our added discussion. Specifically, since altered pore-VSD coupling seems to abolish Slc7a5-mediated inhibition in the Kv1.2’LT’ mutant, we suspect that this effect requires intact pore-VSD coupling (rather than some immobilization of the VSD by Slc7a5 that prevents channel opening).